# Transcriptional coactivation by EHMT2 restricts glucocorticoid-induced insulin resistance in a study with male mice

Rebecca A. Lee [1,2], Maggie Chang [1,2], Nicholas Yiv [2,3], Ariel Tsay [2,3], Sharon Tian [2], Danielle Li [2], Coralie Poulard [4], Michael R. Stallcup[5], Miles A. Pufall [6] & Jen-Chywan Wang [1,2,3] ✉

The classical dogma of glucocorticoid-induced insulin resistance is that it is caused by the transcriptional activation of hepatic gluconeogenic and insulin resistance genes by the glucocorticoid receptor (GR). Here, we find that glucocorticoids also stimulate the expression of insulin-sensitizing genes, such as *Irs2*. The transcriptional coregulator EHMT2 can serve as a transcriptional coactivator or a corepressor. Using male mice that have a defective EHMT2 coactivation function specifically, we show that glucocorticoid-induced *Irs2* transcription is dependent on liver EHMT2's coactivation function and that IRS2 play a key role in mediating the limitation of glucocorticoid-induced insulin resistance by EHMT2's coactivation. Overall, we propose a model in which glucocorticoid-regulated insulin sensitivity is determined by the balance between glucocorticoid-modulated insulin resistance and insulin sensitizing genes, in which EHMT2 coactivation is specifically involved in the latter process.

Glucocorticoids (GC) play a key role in metabolic adaptation during stress conditions, such as fasting and starvation[1]. Because of their potent anti-inflammatory and immunomodulatory activities, GC are also used to treat various inflammatory and autoimmune diseases[2,3]. However, chronic exposure to GC, which can be caused by prolonged stress and long-term GC pharmacotherapy, causes adverse effects that include insulin resistance[4–7]. GC convey their signals through an intracellular glucocorticoid receptor (GR), which is a transcriptional regulator that upon binding to hormones, associates with genomic glucocorticoid response elements (GREs). At the GREs, GR recruits a host of transcriptional coregulators to modulate the transcriptional rate of nearby target genes, which initiate GC-regulated physiological and pharmacological responses. Intriguingly, recent studies have found that certain coregulators participate in the regulation of subsets of GR target genes and

cellular physiological responses to GC[8–14]. Thus, if we can identify coregulators that modulate specific GR actions, they could be attractive targets for improving GC therapy.

EHMT2 (a.k.a. G9A) is one of these coregulators. EHMT2 is a protein methyltransferase that usually forms a heterodimer with EHMT1 (a.k.a. GLP) to convert monomethylated lysine 9 of histone H3 (H3K9me) to dimethylated H3K9 (H3K9me2), a repressive epigenetic mark[15,16]. EHMT2 also directly associates with and coactivates GR[9,12,17,18]. Previous studies showed that EHMT2 acts in synergy with other coactivators, such as GRIP1, P300, and CARM1[9,17,18]. The coactivation function of EHMT2 requires an auto-methylation at the lysine 185 of human EHMT2 (lysine 182 of the mouse counterpart)[12]. This methylation creates a docking site for CBX3 (a.k.a. HP1γ)[19] that is critical for the transcriptional coactivation of EHMT2[12]. Thus, mutating lysine 182 of mouse EHMT2 to arginine (K182R) abolishes

[1]Endocrinology Graduate Program, University of California Berkeley, Berkeley, CA 94720, USA. [2]Department of Nutritional Sciences & Toxicology, University of California Berkeley, Berkeley, CA 94720, USA. [3]Metabolic Biology Graduate Program, University of California Berkeley, Berkeley, CA 94720, USA. [4]Inserm U1052, Centre de Recherche en Cancérologie de Lyon, 28 Rue Laennec, 69000 Lyon, France. [5]Department of Biochemistry and Molecular Medicine, University of Southern California, Los Angeles, CA 90089, USA. [6]Department of Biochemistry and Molecular Biology, University of Iowa Carver College of Medicine, Iowa City, IA 52242, USA. ✉e-mail: walwang@berkeley.edu

the coactivation function of EHMT2 without affecting the corepression function because the methyltransferase activity of EHMT2 remains intact[12]. Interestingly, reducing the expression of EHMT2 in A549 human adenocarcinoma cells only affects the expression of a subset of GR target genes, including both GC-induced and -repressed genes[9,12], which are focused on specific physiological pathways, such as cell migration. In the B-ALL cell line NALM6, EHMT2-dependent GR target genes include those in the cell death pathway[13].

The selective role of EHMT2 in GC response and its ability to serve as both a coactivator and corepressor[15,17,18,20] inspired us to examine its role in GC-regulated metabolic functions in the liver. We first reduced the expression of EHMT2 in mouse liver and found that this worsened GC-induced insulin resistance whereas hypertriglyceridemia and hepatic steatosis induced by GC were not affected. Thus, its role in GR actions is somewhat specific. To distinguish which function of EHMT2 (coactivation or corepression) was involved in GC responses on glucose homeostasis, we created mice that carry a mutation of lysine 182 to arginine in EHMT2 ($Ehmt2^{K182R/K182R}$) that specifically diminishes EHMT2 coactivation but not corepression. To our surprise, $Ehmt2^{K182R/K182R}$ mice had similar phenotypes to hepatic EHMT2 knockdown upon treatment with GC. Following these results, we identified EHMT2 coactivation-dependent GR primary target genes using RNA sequencing and tested whether certain GR primary target genes are involved in GC-regulated insulin sensitivity.

## Results

### Hepatic EHMT2 knockdown exacerbated Dex-induced glucose and insulin intolerance

To analyze the role of EHMT2 in GC regulation of hepatic glucose and lipid metabolism, male C57BL/6J mice (referred as wild type or WT mice in this report) were infected with adenovirus expressing scramble small hairpin RNA (shRNA, Ad-Scr mice) or shRNA targeting mouse $Ehmt2$ (Ad-sh$Ehmt2$) to knockdown EHMT2 in mouse liver (Fig. 1a). Two days later, mice were treated with or without dexamethasone (Dex), a synthetic glucocorticoid, in their drinking water for 7 days. An intraperitoneal glucose tolerance test (IGPTT) was then performed after 16 h fasting. Dex-treated Ad-Scr mice were more glucose tolerant than control Ad-Scr mice without Dex treatment (Fig. 1b). However, Dex treatment caused hyperinsulinemia in control mice (Fig. 1c), which indicates that Dex treatment caused insulin resistance in Ad-Scr mice. In agreement with this assessment, an insulin tolerance test (ITT) showed that Dex-treated control mice did not respond to insulin as well as control mice without Dex treatment. (Fig. 1d). For the ITT, mice were treated with Dex for 7 days and the experiment was performed after 2 h fasting. Overall, these results are consistent with our previous report that Dex treatment for a week causes hyperinsulinemia but not glucose intolerance in C57BL/6J mice[21].

Adenovirus-mediated reduction of EHMT2 expression in the liver did not significantly affect glucose tolerance (Fig. 1b). However, Dex-treated hepatic EHMT2 knockdown mice were significantly more glucose intolerant than Dex-treated control mice and plasma insulin levels

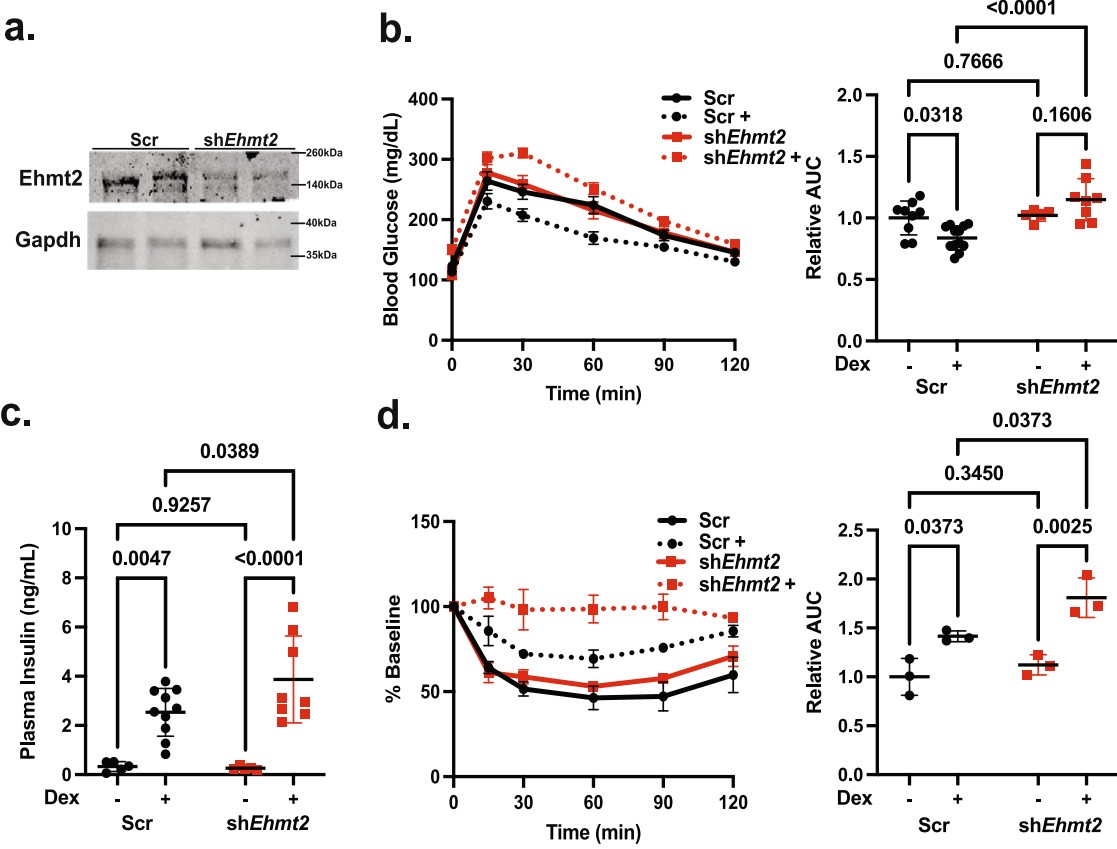

**Fig. 1 | Hepatic EHMT2 knockdown exacerbated Dex-induced glucose and insulin intolerance. a** Western blot showing EHMT2 reduction in shRNA-treated mice from the same experiment repeated twice. **b** WT mice were infected with adenovirus expressing scramble shRNA (Scr) (Black) or $Ehmt2$ shRNA (Red) treated with or without Dex in their drinking water for 1 week and IPGTT was performed (Scr $n = 9$, Scr+ $n = 12$, sh$Ehmt2$ $n = 5$, sh$Ehmt2$+ $n = 8$), **c** plasma insulin levels (Scr $n = 5$, Scr+ $n = 10$, sh$Ehmt2$ $n = 5$, sh$Ehmt2$+ $n = 8$), **d** ITT (performed after 2 hr fasting) $n = 3$ biologically independent mice per group, Statistical tests used were two-way ANOVA with a Holm-Šídák's multiple comparison test, The center lines depict the mean. Error bars represent SEM for the tolerance tests and standard deviation for the rest. Source data are provided as a Source Data file.

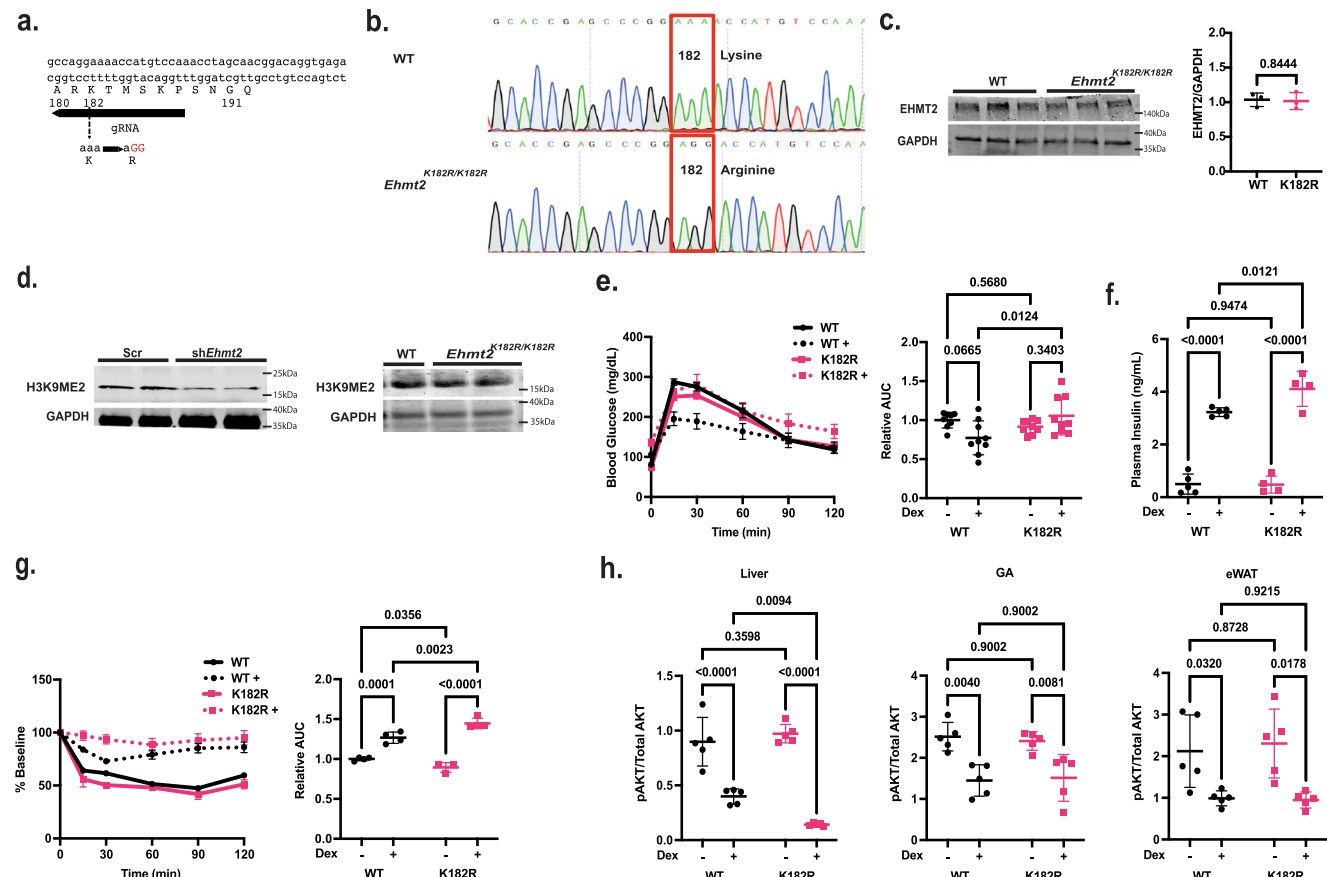

**Fig. 2 | Dex-induced insulin resistance was exacerbated in *Ehmt2^{K182R/K182R}* mice.** **a** gRNA used for creating *Ehmt2^{K182R/K182R}* mice. **b** Sequencing to confirm the aaa to aGG mutation, **c** EHMT2 expression in WT and *Ehmt2^{K182R/K182R}* mouse liver $n = 3$ biologically independent mice. **d** H3K9me2 levels in liver of *Ehmt2^{K182R/K182R}* and hepatic shRNA-*Ehmt2* (sh*Ehmt2*) mice compared to controls repeated twice with biologically independent mice. **e** WT (Black) and *Ehmt2^{K182R/K182R}* (Pink) mice were treated with Dex in their drinking water for 1 week and a IPGTT (WT $n = 8$, WT+ $n = 9$, K182R $n = 9$, K182R+ $n = 9$ biologically independent mice). **f** plasma insulin (WT $n = 5$, WT+ $n = 5$, K182R $n = 4$, K182R+ $n = 4$ biologically independent mice), **g** ITT (WT $n = 4$, WT+ $n = 4$, K182R $n = 3$, K182R+ $n = 4$ biologically independent mice). **h** pAKT/AKT ratios in liver, gastrocnemius muscle (GA), and epididymal white adipose tissue (eWAT) measured with ELISA $n = 5$ biologically independent mice, Statistical tests used were two-way ANOVA with a Holm-Šídák's multiple comparison test, The center lines depict the mean. Error bars represent SEM for the tolerance tests and standard deviation for the rest. Source data are provided as a Source Data file.

were higher (Fig. 1b, c). Dex-treated hepatic EHMT2 knockdown mice also did not respond to insulin as well as mice without Dex treatment (Fig. 1d). Overall, reducing EHMT2 expression in mouse liver exacerbated Dex-induced insulin resistance.

## A Mutation diminishing EHMT2 coactivation but not corepression function exacerbated Dex-induced insulin resistance

EHMT2 can serve as a transcriptional corepressor or a coactivator for GR. Reducing EHMT2 expression compromises both the coactivator and corepressor functions and thus does not distinguish which is responsible for the exacerbated Dex-induced insulin resistance observed above. Inhibiting the methyltransferase activity of EHMT2 also cannot distinguish between whether the corepression or coactivation activity of EHMT2 mediates GC response, as both functions require the methyltransferase activity[12]. Previous studies have shown that GR directly recruits EHMT2 to the GREs of its target genes, where human EHMT2 is automethylated at lysine 185, which creates an anchoring site for another transcriptional coregulator CBX3 to be recruited to GR target genes[12]. The mutation of this lysine residue to arginine significantly attenuates the ability of EHMT2 to coactivate with GR[12]. Notably, this mutation does not affect the methyltransferase activity and therefore the corepressive function of EHMT2 remains intact[12]. We created *Ehmt2* mutant mice (*Ehmt2^{K182R/K182R}*), in which nucleotide sequences encoding lysine 182

(mouse counterpart of lysine 185 in humans) were converted from AAA to AGG (arginine), using CRISPR knock-in technology (Fig. 2a, b). We did not observe any abnormality in *Ehmt2^{K182R/K182R}* mice up to 12 weeks of age. The growth curve, weight gain and food intake of *Ehmt2^{K182R/K182R}* mice with or without Dex treatment were similar to those of WT mice. We performed immunoblotting to monitor EHMT2 and H3K9me2 levels in the liver tissue lysates of WT and *Ehmt2^{K182R/K182R}* mice. We found that hepatic EHMT2 and H3K9me2 levels were comparable between *Ehmt2^{K182R/K182R}* and WT mice (Fig. 2c, d). In contrast, H3K9me2 levels in the liver of hepatic EHMT2 knockdown mice were lower than those of control mice (Fig. 2d). Additionally we performed ChIP for H3K9me2 in the liver of WT, *Ehmt2^{K182R/K182R}*, and EHMT2 knockdown mice. A previous study has performed a whole genome H3K9me2 ChIP-sequencing in mouse liver[22]. One region that contains significant levels of H3K9me2 is chromosome 3 (mm9 chr3:56,379,000–56,380,000). We found that H3K9me2 levels in this region were similar between WT and *Ehmt2^{K182R/K182R}* mice (Supplementary Fig. 1a). But, H3K9me2 levels in this region were lower in EHMT2 knockdown mouse liver (Supplementary Fig. 1a). These results demonstrate that the K182R mutation does not affect the methyltransferase activity of mouse EHMT2 in vivo.

Next, male WT and *Ehmt2^{K182R/K182R}* mice were treated with or without Dex via drinking water for 7 days and an IPGTT was performed.

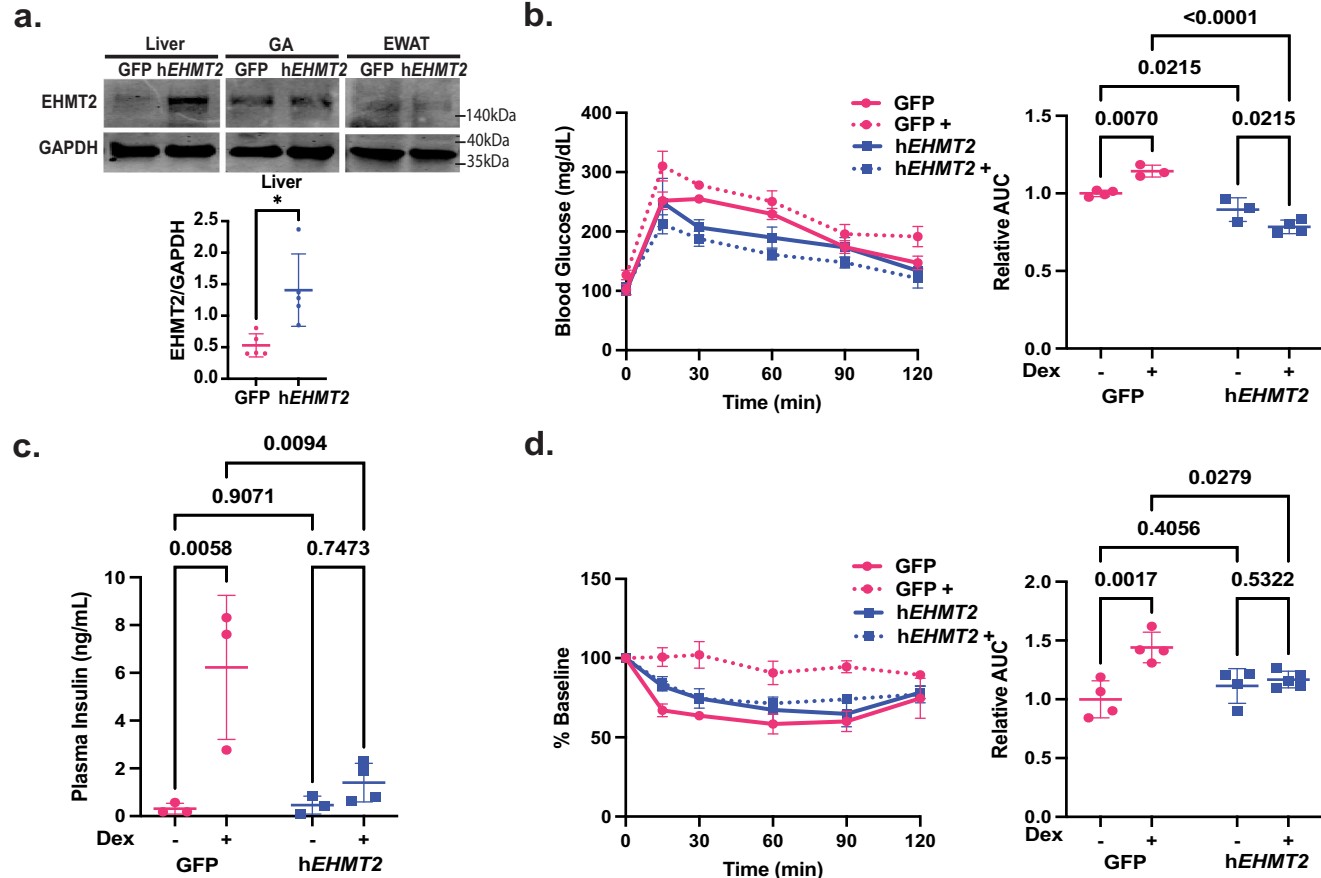

**Fig. 3 | Overexpression of Ehmt2 in liver of *Ehmt2^K182R/K182R* mice improved Dex-induced insulin resistance.** *Ehmt2^K182R/K182R* mice expressing GFP (Pink) or over-expressing human *EHMT2* (Blue). **a** Immunoblot showed EHMT2 overexpression in liver but not gastrocnemius muscle (GA) or epididymal white adipose tissue (eWAT). Blots derived from same experiments and lanes were cropped for rearrangement as pictured. Calculated using ImageJ *n* = 5 biologically independent mice and a two-tailed Student's T-Test, **b** *Ehmt2^K182R/K182R* mice expressing GFP or overexpressing human *EHMT2* were treated with Dex in their drinking water for 1 week and IPGTT was performed, (GFP *n* = 4, GFP+ *n* = 3, h*EHMT2 n* = 3, h*EHMT2+ n* = 4 biologically independent mice) (**c**) plasma insulin (GFP *n* = 3, GFP+ *n* = 3, h*EHMT2 n* = 3, h*EHMT2+ n* = 4 biologically independent mice), **d** ITT (GFP *n* = 4, GFP+ *n* = 4, h*EHMT2 n* = 4, h*EHMT2+ n* = 5 biologically independent mice), Statistical tests used were two-way ANOVA with a Holm-Šídák's multiple comparison test, The center lines depict the mean. Error bars represent SEM for the tolerance tests and standard deviation for the rest. Source data are provided as a Source Data file.

Dex treatment was trending to improve glucose tolerance in WT mice due to hyperinsulinemia (Fig. 2e, f). Without Dex treatment, glucose tolerance, and plasma insulin levels were similar between *Ehmt2^K182R/K182R* and WT mice (Fig. 2e, f). Surprisingly, Dex-treated *Ehmt2^K182R/K182R* mice were more glucose intolerant than Dex-treated WT mice (Fig. 2e) and plasma insulin levels were significantly higher (Fig. 2f). *Ehmt2^K182R/K182R* mice that were treated with Dex for 7 days were also less sensitive to insulin than Dex-treated WT mice as shown by the ITT (Fig. 2g). These observations were similar to those of hepatic EHMT2 knockdown mice presented above. Notably, insulin tolerance was slightly lower in *Ehmt2^K182R/K182R* mice without Dex treatment compared to WT mice without Dex treatment (Fig. 2g).

WT and *Ehmt2^K182R/K182R* mice were treated with or without Dex for 1 week and insulin was injected into these mice and liver, gastrocnemius muscle, and epididymal white adipose tissue were isolated. We then monitored the ratio of phosphorylated protein kinase AKT at serine 473 (pAKT) and total AKT in these tissues via ELISA. PAKT versus total AKT (pAKT/AKT) ratio is an indicator of the activation of insulin signaling because insulin activates AKT through inducing the phosphorylation on serine 473[23]. We found that pAKT/AKT ratios were lower in all three tissues of Dex-treated WT mice compared to untreated mice (Fig. 2h). For *Ehmt2^K182R/K182R* mice treated with Dex, pAKT/AKT ratios were even lower in the liver, but not in the GA and eWAT. Overall, we conclude that the coactivation, not the

corepression activity, of EHMT2 is involved in the regulation of GR-modulated insulin sensitivity and the liver is the major target tissue of EHMT2's coactivation effect.

## Overexpression of EHMT2 in the liver of *Ehmt2^K182R/K182R* mice improved GC-induced insulin intolerance

To test whether the exacerbated GC-induced insulin resistance observed in *Ehmt2^K182R/K182R* mice is due to EHMT2's function in the liver, we infected *Ehmt2^K182R/K182R* mice with adeno-associated virus serotype 8 (AAV8) expressing GFP (AAV8-GFP) or human *EHMT2* (AAV8-h*EHMT2*). Immunoblot confirmed the overexpression of EHMT2 in the liver but not in the gastrocnemius muscle and epididymal white adipose tissue (Fig. 3a). If Dex-induced insulin resistance is improved because of EHMT2 being specifically expressed in the liver, this would validate the key role of hepatic EHMT2's coactivation function in the phenotypes we observed.

Two weeks post-infection, mice were treated with or without Dex in their drinking water for 1 week, an IPGTT was performed, and blood was collected to measure plasma insulin levels. Additionally, an ITT was performed in another set of mice with the same treatment protocol. Dex-treated human EHMT2 overexpressing mice were more glucose tolerant, had lower plasma insulin, and responded to insulin better than those of Dex-treated GFP-expressing mice (Fig. 3b–d). These results indicate that the exacerbated Dex-induced

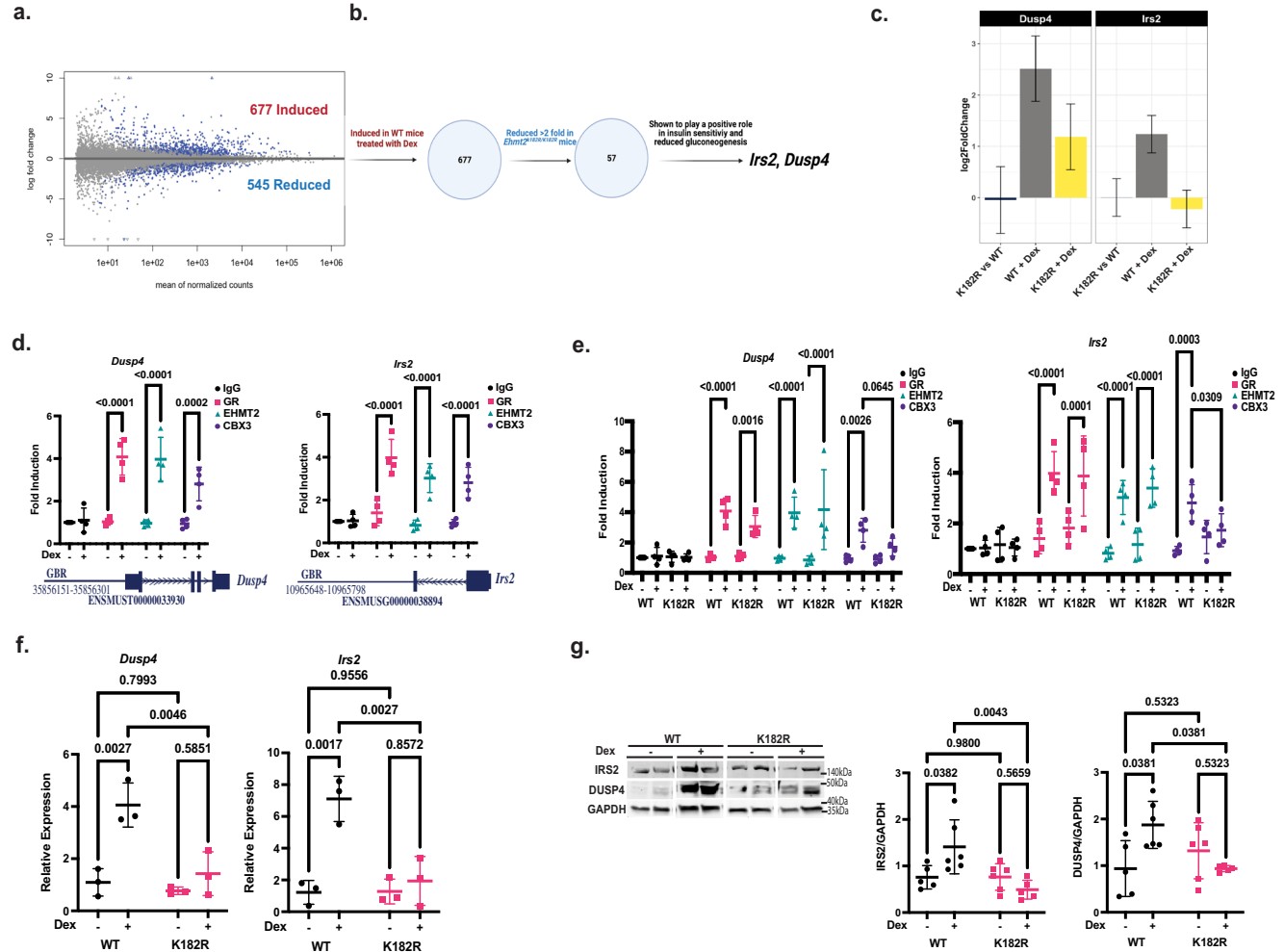

**Fig. 4 | RNA-seq identified EHMT2 coactivation dependent potential GR primary target genes. a** Dex induced and reduced genes in WT mouse livers treated with 2 mg/kg Dex for 5 h via IP injection, **b** Identification of WT Dex-induced genes that the fold induction is reduced in *Ehmt2^{K182R/K182R}* mouse livers, **c** The effect of K182R on Dex regulation of *Dusp4* and *Irs2* from RNA-seq, **d** ChIP in 1 week Dex treated WT mouse liver for GR (Pink), EHMT2 (Green), and CBX3 (Purple) to mouse *Dusp4* (35856151–35856301) and *Irs2* (10965648–10965798) GBR, *n* = 4 biologically independent mice in four independent ChIP experiments. **e** ChIP in 1 week Dex treatment in WT and *Ehmt2^{K182R/K182R}* mouse liver for GR, EHMT2, and CBX3 to mouse *Dusp4* and *Irs2* GBR, *n* = 4 biologically independent mice in four independent ChIP

experiments, using a two-way ANOVA with a Fisher's LSD post hoc test, **f** Gene expression of *Dusp4* and *Irs2* in mouse liver of WT (Black) and K182R (Pink) mice treated with Dex or without Dex *n* = 3 biologically independent mice. **g** Western blot from the same gel (cropped for rearrangement) showing protein levels of IRS2 and DUSP4 in WT and *Ehmt2^{K182R/K182R}* mouse livers normalized to GAPDH using ImageJ (WT- *n* = 5, WT+ *n* = 6, K182R- *n* = 6, K182R+ *n* = 5 biologically independent mice). Statistical tests used were two-way ANOVA with a Holm–Šídák's multiple comparison test, The center lines depict the mean. Error bars represent standard deviation for the rest. Source data are provided as a Source Data file.

insulin resistance observed in *Ehmt2^{K182R/K182R}* mice is mainly attributed to hepatic EHMT2's function. Moreover, the abundance of overexpressed wild type EHMT2 was likely significantly more than that of endogenous K182R proteins (Fig. 3a). Therefore, wild type EHMT2 can overcome the potential dominant negative effect of K182R. Overexpression of EHMT2 improved glucose tolerance but not the insulin response in mice without Dex treatment, and therefore the effect was weaker than those mice treated with Dex (Fig. 3b–d). This is likely because mice without Dex treatment were already insulin sensitive.

## Identification of EHMT2 coactivation function dependent potential GR primary target genes

GC exert their responses through the intracellular GR: a ligand-activated transcription factor. Genes regulated by GR initiate the physiological and/or pathophysiological responses of GC. Thus, a key question is which EHMT2 coactivation-dependent GR target genes promote insulin sensitivity. To identify EHMT2 coactivator-dependent

primary GR target genes in the liver, male WT and *Ehmt2^{K182R/K182R}* mice received intraperitoneal injections of PBS or Dex for a short period of time (5 h) at 5 a.m. in the morning and were euthanized at 10 a.m. Liver RNAs were isolated and RNA-seq was performed. The analysis of RNA-seq is presented in Supplementary Data 1.

The K182R mutation did not cause a large effect overall on gene expression, nor a large effect on Dex regulation of genes. When comparing the K182R mutant mouse liver to WT, only 54 genes changed expression (adjp < 0.01, Supplementary Data 1.C.1.a), with 23 increasing and 31 decreasing in expression. There was no general blunting of regulation by Dex in the livers of K182R mutant mice, which is consistent with previous findings in cell lines that EHMT2 supports Dex regulation of only a minor subset of GR target genes[13,24]. A comparable, if not larger, number of genes were Dex regulated in the livers of mutant mice compared with WT mice (Supplementary Data 1 Fig. S1). Most genes are regulated similarly (Supplementary Data 1 Fig. S1), though some clearly change in their response to Dex. For example, there are 60 genes that are unregulated by Dex in WT

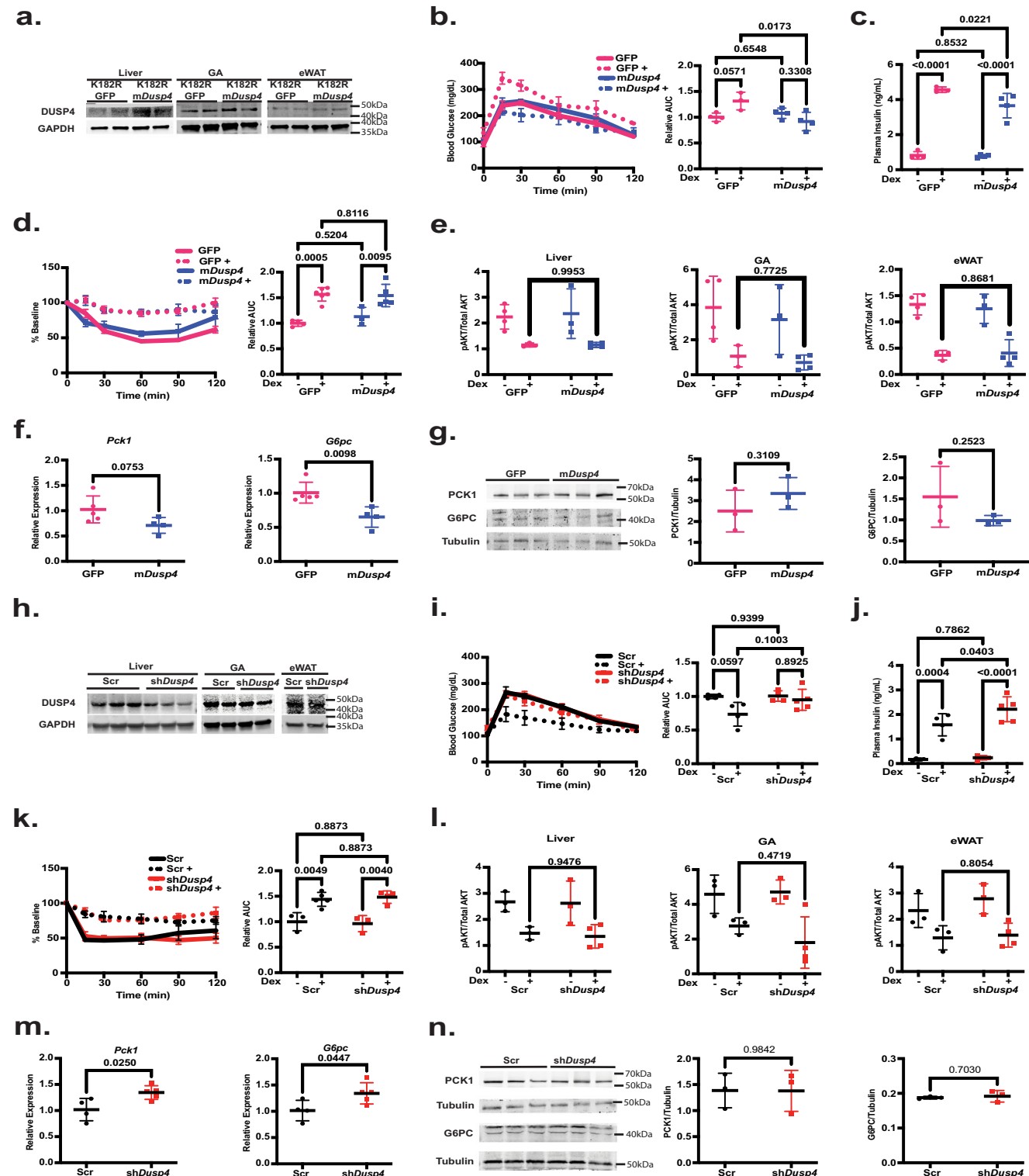

(adjp >0.1) but are strongly activated or repressed in the K182R livers (adjp < 0.01, >2-fold difference). Thus, the effect of K182R does not appear to be solely restricted to genes activated by Dex in WT mouse liver.

Nonetheless, because our previous work indicated that the methylation of K182 of EHMT2 was critical for its coactivation[12], we focused on upregulated genes. In the liver of WT mice, 677 genes were upregulated by Dex (adjp < 0.01, Fig. 4a). These include previously identified glucocorticoid-induced genes, such as *Dusp1*[25], *Sgk1*[26], and *Fkbp5*[27]. Of these 677 genes, the upregulation of 57 genes

was strongly attenuated in the K182R mutant (log2 fold change over 2 fold lower than WT) (Fig. 4b and Supplementary Data 1.D.4). Thus, based on this analysis, the K182R mutation affected only 8.4% (57/677) of Dex-induced genes, highlighting the gene-specific role of EHMT2 coactivation in GR actions. Among these 57 genes, *Irs2* and *Dusp4* (Fig. 4c) have been shown to play a positive role in insulin sensitivity and the reduction of gluconeogenic gene expression[28,29]. *IRS2* has also been found as a glucocorticoid-regulated gene previously[30]. Their roles in GC-regulated insulin sensitivity were examined in the experiments below.

**Fig. 5 | Overexpression of DUSP4 attenuated Dex-induced glucose but not insulin resistance. a** Western blot of DUSP4 shows overexpression in liver but not gastrocnemius muscle (GA) or epididymal white adipose tissue (eWAT), **b** IPGTT in *Ehmt2*[K182R/K182R] mice expressing GFP (Pink) or overexpressing mouse *Dusp4* (m*Dusp4*) (Blue) (GFP n = 4, GFP+ n = 3, m*Dusp4* n = 4, m*Dusp4*+ n = 4 biologically independent mice), **c** plasma insulin (GFP n = 4, GFP+ n = 6, m*Dusp4* n = 3, m*Dusp4*+ n = 4 biologically independent mice), **d** ITT (GFP n = 4, GFP+ n = 3, m*Dusp4* n = 3, m*Dusp4* + n = 5 biologically independent mice), **e** pAKT/AKT ratios in liver, gastrocnemius muscle (GA), and epididymal white adipose tissue (eWAT) measured with ELISA (GFP n = 4, GFP+ n = 3, m*Dusp4* n = 3, m*Dusp4*+ n = 4 biologically independent mice), **f** Gene expression (GFP+ n = 4, m*Dusp4*+ n = 5 biologically independent mice) and **g** western blot n = 3 biologically independent mice for *Ehmt2*[K182R/K182R] mice expressing GFP or overexpressing mouse *Dusp4* treated with Dex, **h** Western blot of DUSP4 shows knockdown in liver but not gastrocnemius muscle or epididymal white adipose tissue (**i**) IPGTT in WT mice with shRNA-Scramble (Black) or *Dusp4* (Red) (Scr n = 4, Scr+ n = 4, sh*Dusp4* n = 4, sh*Dusp4*+ n = 5 biologically independent mice), **j** Plasma Insulin (Scr n = 4, Scr+ n = 4, sh*Dusp4* n = 4, sh*Dusp4*+ n = 5 biologically independent mice), **k** ITT (Scr n = 3, Scr+ n = 5, sh*Dusp4* n = 3, sh*Dusp4*+ n = 4 biologically independent mice), **l** pAKT/AKT ratios in liver, gastrocnemius muscle (GA), and epididymal white adipose tissue (eWAT) measured with ELISA (Scr n = 3, Scr+ n = 3, sh*Dusp4* n = 3, sh*Dusp4*+ n = 4 biologically independent mice), **m** Gene expression (Scr+ n = 4, sh*Dusp4*+ n = 5 biologically independent mice) and **n** Western blot n = 3 for biologically independent mice WT mice with shRNA-Scr or sh*Dusp4* treated with Dex. Statistical tests used were two-way ANOVA with a Holm-Šídák's multiple comparison test, or an unpaired two-tailed t test, The center lines depict the mean. Error bars represent SEM for the tolerance tests and standard deviation for the rest. Source data are provided as a Source Data file.

To confirm whether *Irs2* and *Dusp4* were EHMT2-dependent GR primary target genes, we performed GR and EHMT2 ChIP on the liver of WT mice treated with or without Dex in their drinking water for 1 week to examine whether they were recruited to the GR binding regions (GBRs) of these two genes[31]. A previous study found that one GBR is located near the *Dusp4* gene whereas two GBRs are identified near the *Irs2* gene[31]. Among *Irs2* GBRs, the one located at −42.7 kb from the transcription start site (chromosome 8, between 10965648 and 10965678, based on mm9 assembly) showed a significant GR occupancy in conventional ChIP experiments. Because we have not been able to confirm GR occupancy of another GBR (located at −34.1 kb, between 10974197 and 10974347), we therefore focused on the −42.7 kb GBR. Dex treatment increased the recruitment of GR, EHMT2, and CBX3 to the *Irs2* and *Dusp4* GBRs (Fig. 4d). To examine whether *Ehmt2*[K182R/K182R] mice have impaired Dex-induced recruitment of CBX3 to these GBRs, we also performed liver ChIP in *Ehmt2*[K182R/K182R] for GR, EHMT2, and CBX3. GR and EHMT2 were recruited to the *Irs2* and *Dusp4* GBRs in both WT and *Ehmt2*[K182R/K182R] mice in a Dex-dependent manner. However, Dex-induced CBX3 recruitment was impaired in *Ehmt2*[K182R/K182R] mice (Fig. 4e). Notably, H3K9me2 levels in the GBRs of *Irs2* and *Dusp4* genes were near the background levels and Dex treatment did not affect the H3K9me2 levels (Supplementary Fig. 1b). These results are similar to a previous report, in which glucocorticoid treatment did not affect H3k9me3 levels in the EHMT2 dependent GR target genes that were analyzed[12]. We also monitored *Irs2* and *Dusp4* gene expression in mouse liver in WT and *Ehmt2*[K182R/K182R] mice treated with Dex and observed reduced expression of both in *Ehmt2*[K182R/K182R] mice (Fig. 4f). Additionally we measured IRS2 and DUSP4 protein expression in the liver of WT and *Ehmt2*[K182R/K182R] mice treated with or without Dex. Both IRS2 and DUSP4 proteins were increased by Dex treatment in the liver of WT mice (Fig. 4g). This induction, however, was attenuated in *Ehmt2*[K182R/K182R] mice (Fig. 4g). Overall, these results demonstrated that EHMT2's coactivator function is required for GR to increase the expression of *Irs2* and *Dusp4*.

### Overexpression of DUSP4 in the liver of *Ehmt2*[K182R/K182R] mice ameliorated Dex-induced glucose but not insulin intolerance

We next assessed the roles of DUSP4 and IRS2 in Ehmt2 coactivation-supported GC regulation of insulin sensitivity. We first infected *Ehmt2*[K182R/K182R] mice with AAV8 expressing GFP (AAV8-K182R-GFP) or mouse *Dusp4* (AAV8-K182R-m*Dusp4*). Two weeks after infection, mice were treated with or without Dex in their drinking water for 1 week. Western blot confirmed the overexpression of DUSP4 specifically in liver but not gastrocnemius muscle and epididymal white adipose tissue (Fig. 5a). Dex treatment was trending to induce glucose intolerance and significantly elevated plasma insulin levels (Fig. 5b, c). DUSP4 overexpression did not affect glucose tolerance in control mice but significantly improved glucose intolerance and reduced plasma insulin levels in Dex-treated mice (Fig. 5b, c). DUSP4 overexpression, however, did not affect Dex-induced insulin resistance (Fig. 5d). These

results suggest that DUSP4 reduced hepatic glucose production without affecting insulin signaling. Indeed, AKT activity, measured by pAKT/AKT ratio, was not affected by DUSP4 overexpression in liver, gastrocnemius muscle and epididymal white adipose tissue (Fig. 5e). In contrast, *Pck1* gene expression was trending to be decreased and *G6pc* gene expression was significantly lower in the liver of Dex-treated AAV8-K182R-m*Dusp4* mice compared to that of Dex-treated of AAV8-K182R-GFP mice (Fig. 5f). Pck1 and G6pc protein expression, however, was not significantly decreased in Dex-treated AAV8-K182R-m*Dusp4* mice (Fig. 5g).

To further evaluate the role of DUSP4 in GC-induced insulin resistance, we infected WT mice with AAV8 expressing scramble shRNA (AAV8-Scr) or shRNA targeting *Dusp4* (AAV8-sh*Dusp4*). Two weeks after the infection, mice were treated with or without Dex in their drinking water for 1 week and then an IPGTT was performed. Western blot confirmed knockdown of DUSP4 specifically in liver but not gastrocnemius muscle, and epididymal white adipose tissue (Fig. 5h). Dex treatment did not exacerbate glucose tolerance (Fig. 5i) but caused hyperinsulinemia (Fig. 5i–j). Without Dex treatment, glucose tolerance was similar between AAV8-Scr and AAV8-sh*Dusp4* mice (Fig. 5i). Glucose tolerance was trending to be worsened in Dex treated AAV8-sh*Dusp4* mice than that of Dex-treated AAV8-Scr mice (Fig. 5i). DUSP4 knockdown also was trending to increase plasma insulin levels in Dex treated mice (Fig. 5j). Without Dex, insulin response was not different between AAV8-Scr and AAV8-sh*Dusp4* mice (Fig. 5k). Dex treatment caused insulin resistance in both AAV8-Scr and AAV8-sh*Dusp4* mice (Fig. 5k). Notably, Dex-treated AAV8-sh*Dusp4* mice were not more insulin resistant than Dex-treated AAV8-Scr mice (Fig. 5k). In agreement with these results, Dex-regulated AKT activity in the liver, gastrocnemius muscle, and epididymal white adipose tissue was not affected by hepatic DUSP4 knockdown (Fig. 5l). Expression of *Pck1* and *G6pc* genes were both elevated in Dex-treated AAV8-sh*Dusp4* mice (Fig. 5m). The protein expression of PCK1 and G6PC, however, was not significantly increased in the liver of these mice (Fig. 5n). We performed the pyruvate tolerance test in Dex treated AAV8-Scr and AAV8-sh*Dusp4*. We did not observe a significant difference in the gluconeogenic capacity of these mice, which is agreement with the fact that the protein expression of PCK1 and G6PC was not affected (Supplementary Fig. 2).

### IRS2 plays a key role in mediating EHMT2 coactivation's regulation of Dex response on insulin sensitivity

We next evaluated the role of IRS2 in *Ehmt2*[K182R/K182R] mice with adenovirus expressing GFP (Ad-K182R-GFP) or mouse *Irs2* (Ad-K182R-m*Irs2*). The reason for using adenovirus for *Irs2* overexpression is because mouse and human *Irs2* cDNAs are too large to fit into an AAV vector. IRS2 was specifically overexpressed in the liver but not gastrocnemius muscle and epididymal white adipose tissue as shown through western blot and gene expression (Fig. 6a). Glucose tolerance was similar between Ad-K182R-GFP and Ad-K182R-m*Irs2* mice without

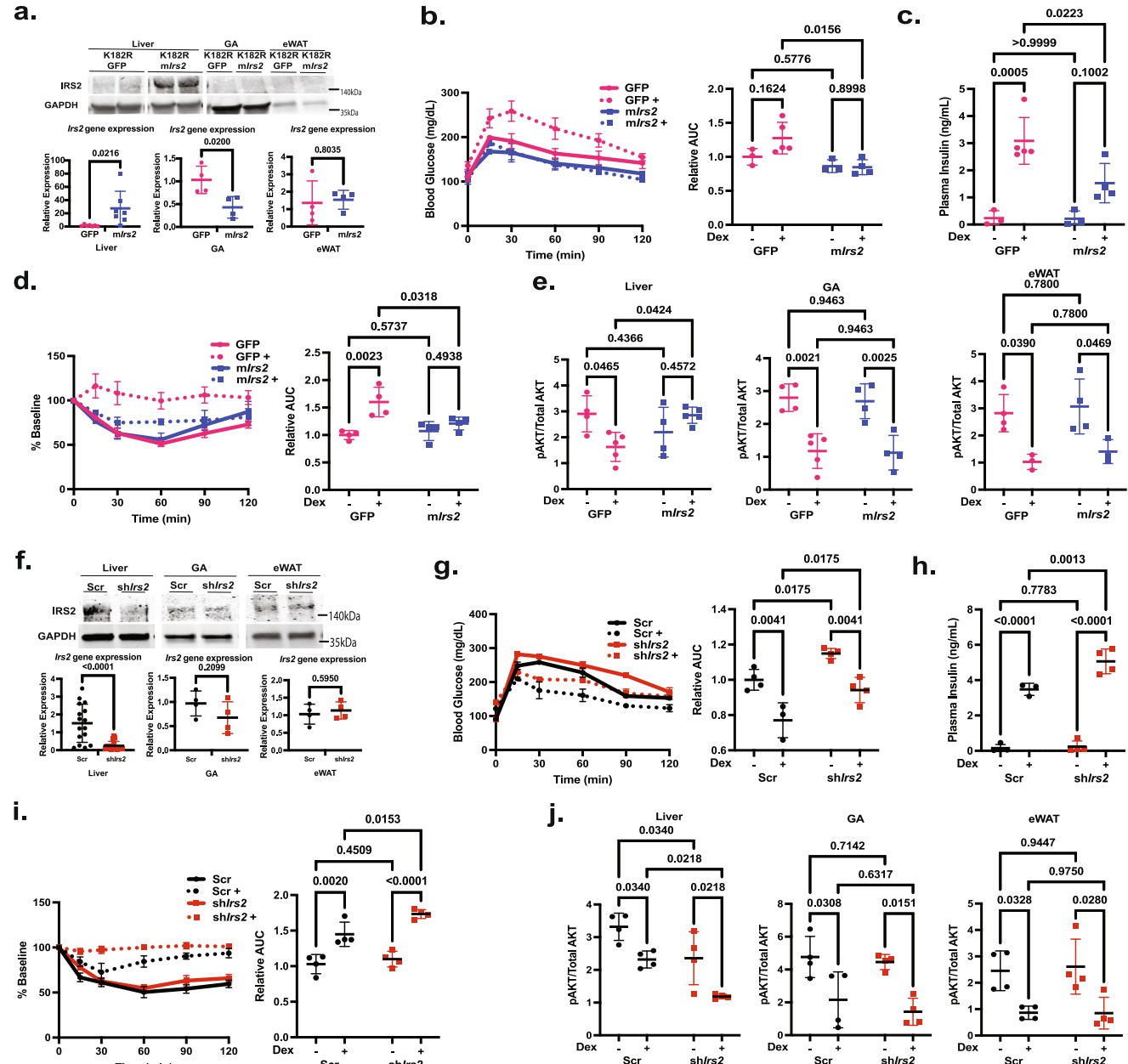

**Fig. 6 | Overexpression of IRS2 improves Dex-induced glucose and insulin intolerance. a** Western blot and gene expression for IRS2 shows overexpression in liver but not gastrocnemius muscle or epidydimal white adipose tissue, gene expression Liver $n = 7$, GA $n = 4$, eWAT $n = 4$ biologically independent mice, **b** IPGTT in *Ehmt2*[K182R/K182R] mice expressing GFP (Pink) or overexpressing mouse *Irs2* (m*Irs2*) (Blue) (GFP $n = 3$, GFP+ $n = 5$, m*Irs2* $n = 3$, m*Irs2*+ $n = 4$ biologically independent mice) (**c**) plasma insulin (GFP $n = 3$, GFP+ $n = 5$, m*Irs2* $n = 3$, m*Irs2*+ $n = 4$ biologically independent mice), **d** ITT n = 4 biologically independent mice, **e** pAKT/AKT ratios in liver, gastrocnemius muscle (GA), and epididymal white adipose tissue (eWAT) measured with ELISA (GFP $n = 4$, GFP+ $n = 5$, m*Irs2* $n = 4$, m*Irs2*+ $n = 5$ biologically independent mice), **f** Western blot and gene expression for Irs2 shows knockdown in liver but not gastrocnemius muscle or epidydimal white adipose tissue. Blots for

each tissue were run separately, but represent mice from the same experiment, gene expression Liver $n = 16$, GA $n = 4$, eWAT $n = 4$ biologically independent mice, **g** IPGTT in *WT* mice with shRNA-Scr (Black) or shRNA-sh*Irs2* (Red) (Scr $n = 4$, Scr+ $n = 3$, sh*Irs2* $n = 4$, sh*Irs2*+ $n = 4$ biologically independent mice), **h** plasma insulin (Scr $n = 4$, Scr+ $n = 3$, sh*Irs2* $n = 4$, sh*Irs2*+ $n = 4$ biologically independent mice), **i** ITT $n = 4$ biologically independent mice, **j** pAKT/AKT ratios in liver, gastrocnemius muscle (GA), and epididymal white adipose tissue (eWAT) measured with ELISA $n = 4$ biologically independent mice. Statistical tests used were two-way ANOVA with a Holm-Šídák's multiple comparison test and an unpaired two-tailed t test, The center lines depict the mean. Error bars represent SEM for the tolerance tests and standard deviation for the rest. Source data are provided as a Source Data file.

Dex treatment (Fig. 6b). Dex treatment did not affect glucose tolerance of Ad-K182R-GFP mice, though plasma insulin was significantly elevated (Fig. 6b, c). However, Dex-treated Ad-K182R-m*Irs2* mice were more glucose tolerant than Dex-treated Ad-K182R-GFP mice and their plasma insulin was also lower (Fig. 6b, c). Insulin tolerance was similar between Ad-K182R-GFP and Ad-K182R-m*Irs2* mice without Dex treatment (Fig. 6d). Not surprisingly, ITT experiments showed that

Dex-treated Ad-K182R-GFP mice did not respond to insulin as well as Ad-K182R-GFP mice that were not treated with Dex (Fig. 6d). However, insulin responses were significantly stronger in Dex-treated Ad-K182R-m*Irs2* mice than those of Dex-treated Ad-K182R-GFP mice (Fig. 6d). PAKT/AKT ratios in the liver, gastrocnemius, and epidydimal white adipose tissue were similar between Ad-K182R-GFP and Ad-K182R-m*Irs2* mice without Dex treatment (Fig. 6e). In Ad-K182R-GFP mice

pAKT/AKT ratios were lower in these three tissues upon Dex treatment (Fig. 6e). Importantly, pAKT/AKT ratios in the liver, but not gastrocnemius muscle and epididymal white adipose tissue, of Dex-treated Ad-K182R-m*Irs2* mice were significantly higher than those of Dex-treated Ad-K182R-GFP mice (Fig. 6e). These results were in agreement with ITT results shown (Fig. 6d).

We also reduced the expression of IRS2 in wild type mice liver to examine its role in Dex-induced insulin resistance. As shown in Fig. 6f through western blot and gene expression, AAV8 expressing *Irs2* shRNA effectively reduced its expression in liver but not gastrocnemius muscle and epididymal white adipose tissue. IRS2 knockdown worsened both basal and Dex-induced glucose intolerance (Fig. 6g). Additionally, IRS2 knockdown increased plasma insulin levels in Dex-treated mice (Fig. 6h). IRS2 knockdown did not affect basal insulin tolerance (Fig. 6i). However, IRS2 knockdown exacerbated the ability of Dex-treated mice to respond to insulin (Fig. 6i). Moreover, IRS2 knockdown further reduced Dex-suppressed AKT activity in the liver but not in the gastrocnemius muscle and the epididymal white adipose tissue (Fig. 6j). These results were similar to those observed in *Ehmt2*$^{K182R/K182R}$ mice (Fig. 2).

Overall, among two EHMT2 coactivation-dependent potential GR primary target genes, IRS2 appears to play a significant role in EHMT2 coactivation modulated GC response on insulin sensitivity. In contrast, DUSP4 is not involved in EHMT2 coactivation-modulated insulin sensitivity.

## Discussion

The classical dogma of GC-induced hepatic insulin resistance is that GC increases the expression of genes encoding proteins promoting hepatic gluconeogenesis, such as *Pck1* and *G6pc*, and inhibit the insulin response (referred to as insulin resistance genes)[4,32,33]. Therefore, when we found that hepatic EHMT2 knockdown exacerbated chronic Dex treatment-induced glucose and insulin intolerance, our first instinct was that EHMT2 acted as a corepressor for GR to activate insulin resistance genes. Surprisingly, our studies found that this is not the case. Similar to hepatic EHMT2 knockdown mice, insulin sensitivity of mice lacking EHMT2's coactivation function was also worsened upon Dex treatment. These results suggest that EHMT2 coactivates with GR to stimulate genes encoding proteins that promote insulin sensitivity

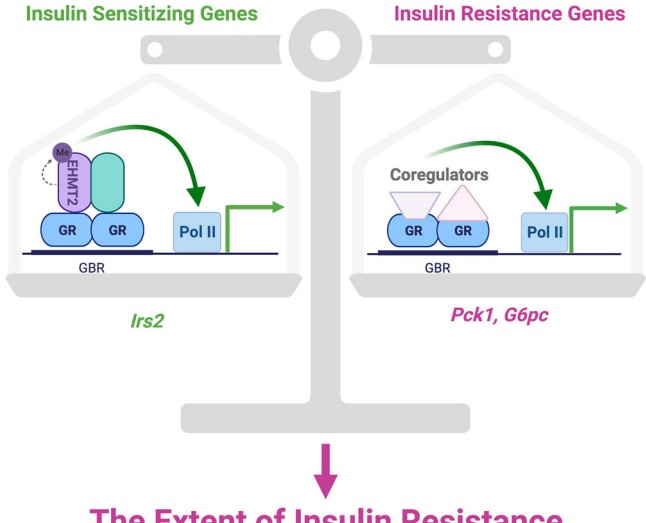

**Insulin Sensitizing Genes** — **Insulin Resistance Genes**

*Irs2* — *Pck1, G6pc*

**The Extent of Insulin Resistance**

**Fig. 7 | Ehmt2's role in balancing chronic glucocorticoid-induced insulin resistance by promoting the transcription of insulin-sensitizing genes.** A model demonstrating the extent of GC-induced insulin resistance to be controlled through the relative expression of GR target genes that play positive and negative roles in glucose homeostasis. The model in Fig. 7 was created with BioRender.com.

or reduce hepatic gluconeogenesis (referred to as insulin-sensitizing genes). This notion that GR could activate insulin-sensitizing genes has not received much attention. RNA-seq found that EHMT2's coactivation function is required for GC to activate a small subset of potential GR primary target genes and that two of these genes, *Irs2* and *Dusp4*, have been previously shown to improve insulin response and/or reduce hepatic gluconeogenic gene expression[28,29]. Both GR and EHMT2 were recruited to previously identified GBRs of these two genes upon Dex treatment. Although *Ehmt2*$^{K182R/K182R}$ mice had worsened insulin sensitivity compared to WT mice upon Dex treatment, overexpression of Irs2 in the liver of *Ehmt2*$^{K182R/K182R}$ mice significantly ameliorated glucose intolerance and insulin resistance in Dex-treated mice. Moreover, reducing IRS2 expression in the liver of WT mice worsened Dex-induced insulin resistance and further diminished Dex-suppressed hepatic AKT activity. These results highlight the role of IRS2 in Dex response on insulin resistance mediated by EHMT2 coactivation function. Notably, a recent study showed that the *Irs2* gene is induced by a glucagon-CREB-PGC1A axis during fasting[28]. While hepatic PGC1A promotes the transcription of gluconeogenic genes to elevate blood glucose levels, the induction of IRS2 primes the liver to respond to insulin to counteract uncontrolled glucose production[28]. A similar concept could be applied to the induction of IRS2 by GC here. Both plasma GC and glucagon are elevated during fasting. The fact that the *Irs2* gene is activated by these two fasting hormones highlights the importance of *Irs2* induction in the control of glucose homeostasis during fasting.

Compared to IRS2, DUSP4 overexpression in *Ehmt2*$^{K182R/K182R}$ mice improved Dex-induced glucose intolerance but had no effect on insulin response. Moreover, reducing the expression of DUSP4 in the liver of WT mice exacerbated Dex-induced glucose intolerance only modestly and had no effect on insulin action. These results demonstrate that DUSP4 is unlikely involved in the EHMT2 coactivation's role in Dex-regulated insulin response. The overexpression and knockdown of DUSP4 modulated the gene expression of *Pck1* and *G6pc* but protein expression of PCK1 and G6PC was not affected. Notably, the *Pck1* and *G6pc* gene expression was not different in the liver of WT and *Ehmt2*$^{K182R/K182R}$ mice under 5 h (RNAseq) or 11 days Dex treatment (Supplementary Fig. 1c). These results further demonstrated that DUSP4 is not involved in the phenotypes of Dex-treated *Ehmt2*$^{K182R/K182R}$ mice. The improvement of Dex-induced glucose tolerance in *Ehmt2*$^{K182R/K182R}$ mice by DUSP4 overexpression is independent of the modulation of PCK1 and G6PC protein expression and could be the result of other function of DUSP4 that remains to be determined.

The ability of GC to induce the transcription of *Irs2* and *Dusp4* revises our view on how GC regulate glucose homeostasis. It is plausible to propose that the extent of GC-induced insulin resistance is determined by the relative expression of GR target genes that play positive and negative roles in glucose homeostasis (Fig. 7). EHMT2 coactivates with GR to specifically activate certain insulin-sensitizing genes but is not involved in GC-stimulated genes encoding gluconeogenic enzymes and proteins that inhibit insulin action, for which other coactivators are required (Fig. 7). A more provocative idea is if there are coactivators that specifically participate in the stimulation of GR-induced insulin resistance genes, then targeting these coactivators could switch GC from insulin resistant to insulin-sensitizing agents.

Other studies have also shown that EHMT2 plays a role in the regulation of glucose homeostasis and insulin sensitivity. A GWAS study has associated a missense mutation in human *EHMT2* gene with type 2 diabetes[34]. It is unclear how this mutation affects EHMT2 activity. Another study in cultured hepatocytes showed that EHMT2 knockdown resulted in downregulation of insulin receptor, phosphorylated AKT and phosphorylated GSK3β. Thus, overexpression of EHMT2 prevents palmitic acid- or glucosamine-induced insulin

resistance by preserving normal insulin signaling. In this study, methyltransferase activity of EHMT2 is attributed for the beneficial effect of EHMT2 on insulin action[35]. Notably, inhibiting methyltransferase activity will reduce both the coactivation and corepression activities of EHMT2.

Among Dex-induced genes, only a small subset of them were affected by the K182R mutation. These results highlighted the specific function of EHMT2 coactivation on GR actions in vivo. However, it is important to note that there are other groups of genes whose expression were affected by this mutation in different ways. There is a group of genes that was not regulated by Dex in the liver of WT mice but was induced by Dex in *Ehmt2^K182R/K182R* mice. In another group of genes, expression was elevated by Dex in the liver of WT mice and further enhanced in *Ehmt2^K182R/K182R* mice. In these two groups of genes, K182R appears to play a negative role on Dex responses. However, we have not confirmed that these genes were directly regulated by EHMT2. We do not exclude the possibility that K182R plays a negative role in GR-regulated gene transcription. A methylated K182 residue recruits CBX3, which can serve as either a transcriptional activator or repressor[19,36]. Finally, there are genes whose expression was reduced by the K182R mutation without Dex treatment. This is not surprising, as EHMT2 associates with other DNA-binding transcription regulators in addition to GR[17,20,37–39].

In summary, we provide two contributions to GC biology in this study. First, we identify EHMT2 as a GR coactivator that specifically participates in stimulating genes that promote insulin sensitivity. The fact that GR activates insulin-sensitizing genes leads us to propose a refined model in which the balance between GR-regulated insulin resistance-promoting and insulin-sensitizing genes determine the effects of GC on glucose homeostasis (Fig. 7). *Irs2* is likely not the only glucocorticoid-regulated gene that has a positive role in regulating glucose homeostasis. Identifying other glucocorticoid-regulated genes that promote insulin sensitivity is important to our understanding of the complex regulation of glucose homeostasis by GC. Another key finding in this study is that certain coregulators, such as EHMT2 here, participate in regulating a specific subset of GR primary target genes to be selectively involved in distinct aspects of GC actions. Thus, such coregulators could be the potential targets to dissociate GC responses, which will provide innovative approaches to improve glucocorticoid pharmacology.

## Methods
### Animals
*Ehmt2* mutant mice (*Ehmt2^K182R/K182R*) were created with the help of The Washington University Genome Engineering and iPSC Center which performed in vitro validation of the gRNA needed for the CRISPR experiments in N2a cells. We chose gRNA (AGGTTTGGA CATGGTTTTCCNGG) that not only mutated K182 to R (amino acid numbers are based on mouse *Ehmt2* isoform b, NP_671493.1) but also had the least off-target sites. Once the gRNA needed to create this mutation was identified in vitro, it was injected into zygotes (C57BL/6J) along with the Cas9 protein. The blastocysts derived from the injected zygotes were implanted into foster mice. These steps were carried out at the Gene Targeting Facility of the Cancer Research Institute at UC Berkeley. Overall, nine founders were born, and genotyping showed that eight of them had homozygous mutations that convert AAA to AGG. We bred these homozygous *Ehmt2^K182R/K182R* mice for our preliminary studies. C57BL/6J mice from Jackson laboratories were used as WT controls. Mice were co-housed or individually housed in a temperature-controlled room of ~22 °C with 30–70% humidity in ventilated cages with a 12 h light and dark cycle. Cages include Sanichip bedding along with a cotton Nestlet and a 4 g pick of crinkled paper. Mice were cohoused and were fed ad libitum a diet of PicoLab Rodent diet 5053 which contains 20% protein, crude fat 4.5%, and Fiber 6.0%. For the following experiments, randomly assigned male mice

8–12 weeks old were used. All experiments were approved by the University of California, Berkeley (AUP-2014-07-6617).

### Adenovirus/AAV
Adenovirus was purchased from Vector Biolabs (Malvern, PA). Virus was diluted in sterile PBS and injected via tail vein at $2 \times 10^9$ PFU per male mouse (shRNA-Scr (#1122), shRNA-mouse *Ehmt2* (built to order CCGG-CCGAGAGAGTTCATAGCTCTTCTCGAGAAGAGCTATGAACTCT CTCGG-TTTTT), GFP (#1060), mouse Irs2 (#ADV-262384)). AAV8s were purchased from Vector Biolabs (Malvern, PA). Virus was diluted in sterile PBS and was injected via tail vein at $3 \times 10^{11}$GC per male mouse (shRNA-Scr, shRNA-mouse *Dusp4*(#shAAV-257549 5′-CCGGGCTGATG AACCGGGATGAGAA-CTCGAG-TTCTCATCCCGGTTCATCAGC-TTTTT G-3′ and a targeting sequence of GCTGATGAACCGGGATGAGAA), mouse *Dusp4*(#AAV-257549), human *EHMT2* (#AAV-207689), GFP (#7061), shRNA-mouse *Irs2* (5′-CCGG-TCATGTCCCTTGACGAGTATGC TCGAGCATACTCGTCAAGGGACATGA-TTTTT-3′)).

### Dexamethasone water supplementation
Male mice were treated with ~2 mg/kg bodyweight Dexamethasone (Dex). Dexamethasone sodium phosphate (Sigma, St. Louis, MO, PHR 1768) was diluted in water to a concentration of 15.5 mg/l. Dexamethasone sodium phosphate has a molecular weight of 516.4 g/mol and dexamethasone has a molecular weight of 392.47 g/mol. There is 760 mg of dexamethasone per gram of dexamethasone sodium phosphate powder. We prepared to drink water that contains 0.0155 g of dexamethasone per liter and based our calculations on the estimate that a 30 g mouse drinks ~3.5 ml of water per day. Mice treated with adenovirus were treated with Dex two days after injection. Mice treated with AAV were treated with Dex 2 weeks after injection.

### Glucose, insulin, and pyruvate tolerance tests
The following tests were performed after 1 week of Dex treatment in male mice 9–13 weeks old at the time of the test. For glucose and pyruvate tolerance tests, mice were fasted for 16 h. For insulin tolerance tests, mice fasted for 2 h. Mice were transferred to a clean cage and food was removed. Mice were freely able to consume water. Fasting blood glucose measurements and weights were taken. For GTTs, 1 g/kg D-glucose (Sigma, St. Louis, MO, 50-99-7) dissolved in PBS was used. For ITTs, 1 U/kg of insulin was injected (Sigma, St. Louis, MO, I0516-5ml). For PTTs, 2 g/kg Sodium Pyruvate (Sigma, St. Louis, MO, P2256) dissolved in PBS was injected. Blood glucose levels were taken on a glucometer (Contour, Bayer, Parsippany, NJ) at 0, 15, 30, 60, 90, and 120 min.

### Blood and tissue collection
Fasting Blood was collected in EDTA-coated blood collection tubes (Sarstedt, Newton, NC, 16.44.100). Terminal blood collection was performed after euthanasia via cardiac puncture. Blood was put into heparin-coated tubes and was centrifuged at 14,000 *g* 4 °C for 10 min. Plasma was collected and stored at −80 °C until analysis. Tissues were snap-frozen in liquid nitrogen and were stored at −80 °C until analysis.

### Insulin and pAKT/AKT ELISA
Plasma Insulin levels were examined using an ultra-sensitive mouse insulin ELISA kit (Crystal Chem Inc, Downers Grove, IL, Cat No 90080) on plasma collected in EDTA-coated blood collection tubes on mice fasted for 16 h. The pAKT and AKT levels were measured using the AKT (total) ELISA kit (Invitrogen, Carlsbad, CA 85-86046-11). Tissues were homogenized in cell lysis buffer using 50 mg pieces with a BeadBug 6 Homogenizer (Benchmark Scientific, Sayreville, NJ). Homogenized lysates were centrifuged at 14,000 *g* for 15 min 4 °C. Supernatant was transferred to fresh tubes and protein levels were determined with a BCA assay (Thermo Fisher, Waltham, MA, 23225) according to manufacturer's instructions.

## RNA-seq

Twelve experiments were performed consisting of three biological replicates of WT + vehicle, WT + Dex, $Ehmt2^{K182R/K182R}$ + vehicle, and $Ehmt2^{K182R/K182R}$ + Dex. Male mice were intraperitoneally injected with 2 mg/kg Dex at 5 a.m. and livers were collected at 10 a.m. Liver RNA was extracted using Zymo's Direct-zol RNA microcrep kit. Samples were then sent to BGI Americas for RNA sequencing. Total RNA underwent sample QC using the Agilent 2100 Bio analyzer for total RNA sample QC: RNA concentration, RIN value, 28S/18S and fragment length distribution. Reads from BGI were first trimmed to remove adapters (TrimGalore! v0.6.6: trim_galore –gzip sample_1.fastq.gz), then mapped to the ensemble transcriptome (Mus_musculus.GRCm38.cdna.all.fa.gz) using Salmon (v1.3.0: salmon quant -i GRCm38_index -l A -r sample_1_trimmed.fq.gz -p 4 --validateMappings --seqBias --gcBias --numBootstraps 50 -o quants/sample_1_quant). Count tables were imported into R using tximeta (v 1.9.3) using the gene model Mus_musculus.GRCm38.101.gtf.gz, with differential expression performed using DESeq2 (v 1.30.1, and fit to the model -dex + mutant + dex:mutant). A more detailed accounting of the processing in R is included as an Rmarkdown file in Supplementary Data 1. The results of all RNA-seq genes analyzed are listed in Supplementary Data 2. Fifty-seven genes differentially regulated by dex in the liver of WT and $Ehmt2^{K182R/K182R}$ mice are highlighted by yellow.

## Western blot

Tissues were homogenized in RIPA buffer with protease and phosphatase inhibitors using a BeadBug 6 Homogenizer (Benchmark Scientific). Homogenized lysates were centrifuged at 14,000 $g$ for 15 min 4 °C. Supernatant was transferred to fresh tubes and protein levels were determined with a BCA assay according to manufacturer's instructions. Approximately 30 μg of protein was mixed with 1x NuPAGE LDS Sample buffer (ThermoFisher, Waltham, MA NP007) and 1× NuPAGE sample reducing agent (Thermofisher, Waltham, MA, NP009). They were boiled for 5 min and were applied to SDS Page. The following antibodies were used: GAPDH 1:1000 (Proteintech, Rosemont, IL, 10494-1-AP), EHMT2 1:1000 (Sigma, St. Louis, MO, SAB 2100657), DUSP4 1:500 (Cell Signaling, Danvers, MA, 5149), H3k9ME2 1:1000 (Abcam, Cambridge UK, ab32521), IRS2 1:1000 (Cell Signaling, Danvers, MA, 3089 S), PCK1 1:2000 (Proteintech, Rosemont, IL, 16754-1-AP), G6PC 1:500 (Proteintech, Rosemont, IL, 22169-1-AP), Alpha Tubulin 1:1000 (Proteintech, Rosemont, IL, 11224-1-AP). Developed on Li-Cor using 1:10000 of mouse anti rabbit (Li-Cor, Lincoln, NE, 926032211) or goat anti mouse 800 (LiCor Lincoln, NE, 926-32210).

## Gene expression

Total RNA was isolated from liver tissues using TRIzol reagent (Invitrogen, Carlsbad, CA, 15596018). Reverse transcription was performed as following: 0.5 μg of total RNA, 4 μl of 2.5 mM dNTP, and 2 μl of 15 μM random primers (New England Biolabs, Ipswich, MA, S1254S) were mixed at a volume of 16 μl, and incubated at 70 °C for 5 min. Then, a 4 μl cocktail containing 25 units of Moloney Murine Leukemia Virus (M-MuLV) Reverse Transcriptase (New England Biolabs, Ipswich, MA, M0253S), ten units of RNasin Plus (Promega, Madison, WI, N261B) and 2 μl of 10x M-MuLV Reverse Transcriptase Reaction Buffer (New England Biolabs, Ipswich, MA, B0253S) was added, and samples were incubated at 42 °C for 1 h and then at 95 °C for 5 min. The cDNA was diluted and used for real-time quantitative PCR (qPCR) using the Power Eva qPCR SuperMix Kit (Biochain, Newark, CA, K5057400), following manufacturer's protocol. The qPCR was performed on the StepOne PCR System (Applied Biosystems, Foster City, CA) and analyzed with the ΔΔ-Ct method, as supplied by the manufacturer (Applied Biosystems, Foster City, CA). $Rpl19$ gene expression was used for internal normalization. Primers are listed in Supplementary Table 1.

## Liver ChIP

The protocol of liver ChIP was previously reported[40] with some revisions. Briefly, 300 mg pieces of liver were minced in 10 ml 1× SCC buffer (20× SSC: 175.3 g NaCl, 88.2 g Sodium citrate in 1 L water pH7, 1× SSC made with diluting with 10 mM Tris HCl pH 7.5) on ice, then was centrifuged at 4000 $g$ 3 min 4 °C. Supernatant was discarded, and liver pieces were resuspended in 20 mL of PBS and were crosslinked with 1% formaldehyde for 10 min with rocking at room temperature. Crosslinking was quenched with 125 mM of glycine for 3 min. Samples were then centrifuged at 4000 $g$ 3 min 4 °C and pellets were resuspended in PBS with protease inhibitor. Samples were then centrifuged again at 4000 $g$ 3 min 4 °C and supernatant was discarded. Samples were resuspended in 5 ml of hypotonic buffer (10 mM Hepes pH 7.9, 1.5 mM MgCl$_2$, 10 mM KCl, 0.2% NPD, 1 mM EDTA, 5% Sucrose, protease inhibitor, spermine, spermidine) and were incubated on ice for 5 min and were then transferred to a 7 mL dounce homogenizer and were hit with 10 strikes on ice. Nuclei solution were mounted gently on 5 mL cushion buffer (10 mM Tris HCl pH 7.5, 15 mM NaCl, 60 mM KCl, 1 mM EDTA, 10% Sucrose, protease inhibitor, spermine, spermidine). Samples were then centrifuged at 4000 $g$ 4 °C 3 min and the supernatant was removed. Pellet was resuspended in sonication buffer (50 mM Tris HCl pH 8, 2 mM EDTA) and was sonicated at 60% amplitude for 10-s bursts, 40 s rests for 5 min total bursts for the GR, EHMT2, CBX3 ChIPs. H3k9ME2 ChIPs were sonicated at 60% amplitude for 15 s bursts, 50 s rests for 4 min 10 s total bursts. Sonicated sample was spun at 14,000 $g$ for 10 min 4 °C and supernatant was collected and used for setting up the immunoprecipitation. Samples were incubated with 4 μg overnight with rotation at 4 °C of the following antibodies: IgG (GeneScript, Piscataway, NJ, A01008), GR (IA-1; a polyclonal rabbit antibody raised against human GR amino acids 84–112 QPDLSKAVSLSMGLYMGE-TETKVMGNDLG), EHMT2 (Abcam, Cambridge, UK, 40542), CBX3 (Abcam, Cambridge, UK, 10480), H3k9ME2 (Abcam, Cambridge, UK, 32521). 40 μl of 25% protein A/G sepharose beads were added and incubated for 2 h at 4 °C with rotating. Beads were washed with TSE I (0.1% SDS, 1% Triton X-100, 2 mM EDTA, 20 mM Tris HCl pH 8, 150 nM NaCl) once rotating 5 min 4 °C and centrifuged 3000 $g$ 1 min, TSE II (0.1% SDS, 1% Triton X-100, 2 mM EDTA, 20 mM Tris HCl pH 8, 500 mM NaCl) once, TSE III (0.25 M LiCl, 1% NP40, 1% Na-deoxycholate, 1 mM EDTA, 10 mM Tris HCl pH 8) once, TE (10 mM Tris HCl pH 8, 1 mM EDTA) twice. Afterward, all supernatant was removed, and samples were resuspended in 400 μl Elution Buffer (100 mM NaHCO3, 1% SDS) to the IPs and to the input tubes up to 400 μl. Samples then were rotated 1 h at room temperature. After elution NaCl was added to a final concentration of 200 mM and each tube and were incubated at 65 °C overnight. 8 μl 0.5 M EDTA, 16 μl 1 M Tris HCl pH 6.5, 1.5 μl Proteinase K (Thermo Fisher, Waltham, MA, EO0491), and 1 μl RNase A (Thermo Fisher, Waltham, MA, EN0531) were added to each tube and were incubated at 55 °C for 1 h. Samples were then subjected to PCR cleanup using Qiagen's PCR cleanup kit (Qiagen, Hilden, Germany, 28104)) and were eluted with 40 μl of autoclaved water. The qPCR was performed on the StepOne PCR System (Applied Biosystems, Foster City, CA) and analyzed with the ΔΔ-Ct method, as supplied by the manufacturer (Applied Biosystems, Foster City, CA). H3K9ME2 ChIP primers were designed from chr3:56,379,000–56,380,000[22], and GR[31], EHMT2, CBX3 ChIP primers for $Dusp4$ and $Irs2$ were designed using the mm9 assembly. Primers are listed in Supplementary Table 1.

## Statistics and reproducibility

All experiments were performed at least two or three times. Data were analyzed using Prism version 9 software (Graphpad). Data are expressed as standard deviation (S.D.) or SEM for each group with the center line depicting the mean. Comparisons were analyzed as stated in the figure legends using an unpaired two-sided t test, a one-way ANOVA, or a two-way ANOVA depending on the number of independent variables.

Post-hoc Holm-Šídák's multiple comparison test or Fisher's LSD tests were performed. RNA-seq data were analyzed as described above.

### Reporting summary

Further information on research design is available in the Nature Portfolio Reporting Summary linked to this article.

## Data availability

The RNA-seq data generated in this study have been deposited in the NCBI Gene Expression Omnibus database under accession code GSE179180. The remaining data generated in this study are provided in the Supplementary Data 1 and 2/Source Data file. Source data are provided in this paper.

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

## Acknowledgements
This work was supported by the NIH R01 DK113019 and 124866. J.C.W. is a member of the UCSF Liver Center (NIH P30 DK026743).

## Author contributions
This study was planned and conceptualized by R.A.L., C.P., M.R.S., and J.C.W. R.A.L. and J.C.W. designed and supervised the experiments. R.A.L., M.C., N.Y., A.T., S.T., and D.L. executed the experiments. R.A.L. and M.A.P. analyzed the data, J.C.W., R.A.L., M.A.P, C.P., and M.R.S. wrote the paper.

## Competing interests
The authors declare no competing interests.
