## [Peer Review File · Nature Communications]

Ehmt2's Transcriptional Coactivation Restricts Glucocorticoid-Induced Insulin ResistanceREVIEWER COMMENTS

Reviewer #1 (Remarks to the Author):

This manuscript from the Wang lab aims to shed new light on the mechanisms of glucocorticoid-induced insulin resistance and to describe a novel role for GR in the activation of insulin-sensitizing genes. The authors focus on the putative coregulator Ehmt2 as potential mediator of Dex-induced insulin-resistance in the liver.

The authors show evidence that supports a role of Ehmt2 in hepatic metabolic function: Temporary hepatic knockdown (Ad-sh experiments) exacerbates Dex-induced glucose intolerance, hyperinsulinemia and insulin tolerance, but at the same time doesn't affect lipid homeostasis.

These actions look to be dependent on its coactivator function, as shown by the generation and use of knock-in mice carrying a mutation that specifically impairs the ability of coactivation (K182R). Rescue experiments by using AAV-hEHMT2 overexpression show that the effect on glucose metabolism is liver specific.

Although, in the RNA-seq study, the Dex response is only slightly affected (~8% of induced genes), the authors identified *Irs2* and *Dusp4* as possible targets mediating insulin sensitivity. GR and Ehmt2 recruitment seem to be impaired at GBRs of these loci in the mutant mice, and protein expression fails to increase after Dex treatment.

Further experiments include the transient overexpression and knockdown of the 2 candidate genes. The experiments show involvement of *Irs2* in the modulation of insulin sensitivity. *Dusp4* activation may modulate the Dex-dependent gluconeogenic program.

Although a link between Ehmt2 and GC regulation of glucose homeostasis appears legitimate, it is not clear if the described functions operate during more physiological conditions. Glucocorticoid hormones are secreted during fasting, and their effect on glucose metabolism is particularly evident at this stage. Can the authors elucidate the role of Ehmt2 during the fasting response (24h and 48h)? It would be also insightful to investigate what happens to the mice during re-feeding following fasting, to test for impairments in insulin action.

A validation at protein level of *Dusp4* and *Irs2* downregulation in K182R has been done. Since the regulation appears to happen transcriptionally, it is necessary to confirm the downregulation by qRT-PCR in the mouse cohorts. Also, a visual representation of the peak locations used for ChIP-qPCR will be useful.

Part of the mechanism is explained by *Dusp4* acting on "insulin resistance genes". The authors show that *Dusp4* can indirectly modulate *Pck1* and *G6pc* expression - I found this link to be rather weak, as actually these two genes are not differentially regulated in the mutant mice compared to WT upon DEX. How do the authors explain causality then? Also, what's the protein level of *Pck1* and *G6pc* in *Dusp4* KD and OE?

Minor comments:

- The type of DEX treatment in Fig 1 and 3 is assumed to be in drinking water, but is not specified.
- Similarly, the DEX 5h treatment is not described in the methods. Has it been done by IP injections?

Which dose?

- What's the level of expression of mutant K182R protein? Is this comparable to the WT form?
- Lane 177 a bracket is missing.
- Lane 248, reference to Fig 5i not 5h.
- The term 'revolutionary' is absolutely inappropriate in the abstract.

Overall, this manuscript includes a lot of in vivo data, and the findings are potentially interesting to the field. However, several shortcomings unfortunately prevent publication in its current form. Most importantly, the evidence that Ehmt2 is a direct GR coactivator (in the molecular sense), and that the effects of Ehmt2 loss of function on GR target gene regulation are direct, need to be solidified and expanded.

Reviewer #2 (Remarks to the Author):

The manuscript by Lee et al. examines the mechanism of GR-dependent transcriptional regulation (activation vs repression) in mediating glucocorticoid-induced glucose intolerance and insulin resistance. The authors show that the GR co-regulatory protein, Ehmt2, which is involved in both gene activation and repression, when eliminated in the liver in mice results in more glucose intolerance and insulin resistance upon chronic glucocorticoid treatment without affecting triglyceride synthesis, thus linking Ehmt2 to glucose homeostasis. To determine whether this is a result of a defect in GR-mediated gene activation vs repression, they engineered a mouse to express a single amino acid substitution in Ehmt2 (K182R) that blocks the co-activator function but retains co-repressor function. The authors use this novel model to demonstrate that this resembles the loss of function of Ehmt2, suggesting that the coactivation function of Ehmt2 is important in controlling glucose homeostasis in mice in vivo. Importantly they show that overexpression of wild type Ehmt2 in the liver of mice in the context of the Ehmt2 K182R mutation restores glucose hemostasis, indicating that the liver is the major tissue involved in the glucose homeostasis by GR and Ehmt2. They go on to identify potential GR targets involved in glucose homeostasis by RNA seq from livers of WT vs Ehmt2 mutant mice. They identify DUSP4 and IRS2 as direct GR targets of GR-Ehmt2 by ChIP. Whereas overexpression of DUSP4 in the liver improved glucose tolerance but not insulin insensitivity, IRS2 overexpression improves both glucose and insulin tolerance in the face of chronic glucocorticoid treatment. Overall the data are compelling and are consistent with the interpretation that the coactivation function of Ehmt2 in the liver is the key determinant of glucocorticoid-induced glucose intolerance and insulin resistance. This represents a

novel and important finding.

However there are a few areas of concern. One is that since the Ehmt2 mutant (K182R) mimics Ehmt2 depletion, it begs the question does the Ehmt2 mutant protein (and by association GR) occupy the DUSP4 and IRS2 promoters and fail to coactivate, or does Ehmt2 K182R fail to occupy the DUSP4 and IRS2 regulatory regions resulting in reduced expression? Perhaps this has already been demonstrated by CHIP in the authors previous reports.

It would seem from the Ehmt2 complementation study in the liver (which is a very nice experiment) that WT Ehmt2 protein wins out with respect to gene expression, whereas one might posit if the Ehmt2 K182 mutant occupies DNA, then it would act as a dominant negative and block WT EhMT2 activity. However, this does not seem to be the case. It might be worth noting this in the discussion. Is this because WT Ehmt2 is more abundant than Ehmt2 K182 and able to compete effectively to restore gene expression?

The effects of IRS2 depletion in the liver by shRNA and subsequent effects on insulin resistance is difficult to discern from Fig 6i? The curves of ITT in the shRNA IRS2 knock down appear superimposable, yet the AUC seem quite different? Not sure how to reconcile this?

The discussion seems overly speculative: whereas mentioning the GWAS study linking Ehmt2 to type 2 diabetes seems fitting, the speculation about the impact of Ehmt2 T183 phosphorylation seems too hypothetical. That said, the idea that GR might be promoting the effects of insulin is really interesting, and makes us rethink the dogma about how GR effects insulin signaling.

Reviewer #3 (Remarks to the Author):

The authors claim that Ehmt2 co-activates *Irs2*, an insulin-sensitizing gene, along with the glucocorticoid receptor under glucocorticoid treatment. Their data showed that Dex-induced insulin resistance was exacerbated in sh-RNA-treated mice targeting Ehmt2 and Ehmt2(K182R/K182R) mice, and that the latter was ameliorated by overexpression of human EHMT2 or mouse *Irs2*. Their data showing a correlation between the K182R mutant of Ehmt2 and Dex-induced insulin resistance is interesting and convincing. However, it is still unclear whether the K182R mutation is directly related to the activation of GR-dependent transcription. In addition, considering the site-specific regulations of transcription by histone modifications, the immunoblot analysis of liver tissue extract using anti-H3K9me2 is insufficient to conclude that "K182R mutation does not affect the methyltransferase activity of mouse Ehmt2 in vivo". Therefore, the authors should conduct additional experiments to clarify the mechanism by which Ehmt2 mediates Dex-induced insulin resistance.

1. It is unclear whether the co-activation of GR target genes by Ehmt2 occurs by a mechanism

independent of histone H3K9 methylation. In Figure 4d, ChIP-qPCR should be performed in both WT and Ehmt2(K182R/K182R) mice using the anti-H3K9me2 antibody in addition to anti-GR and anti-Ehmt2 antibodies. Furthermore, it is not clear whether Ehmt2 is recruited by GR and methylated to activate gene expression in mouse liver as reported in the previous study used cell culture models by Poulard et al. (Ref. #12). It should be confirmed whether Cbx3 is recruited to GR target genes in mouse liver in a methylated Ehmt2-dependent manner. If not, what is the mechanism by which GR is activated by methylated Ehmt2 in mouse liver?

2. The manuscript on page 12 does not match the numbering of the figures (Figure 5h and following). In some places, it is not possible to guess which part of the figure is being discussed, which makes it difficult to evaluate the content.

3. The intensity of the immunoblot analysis of Irs2 in GA and eWAT in Fig. 6f is too weak to confirm that there is no significant reduction by sh-Irs2. mRNA expression should alternatively be presented.

4. In relation to the regulation of hepatic gluconeogenesis by Dusp4, the authors should present the expression of the gluconeogenic genes in the liver of Ehmt2(K182R/K182R) mice.

REVIEWER COMMENTS

Reviewer #1 (Remarks to the Author):

This manuscript from the Wang lab aims to shed new light on the mechanisms of glucocorticoid-induced insulin resistance and to describe a novel role for GR in the activation of insulin-sensitizing genes. The authors focus on the putative coregulator Ehmt2 as potential mediator of Dex-induced insulin-resistance in the liver.

The authors show evidence that supports a role of Ehmt2 in hepatic metabolic function: Temporary hepatic knockdown (Ad-sh experiments) exacerbates Dex-induced glucose intolerance, hyperinsulinemia and insulin tolerance, but at the same time doesn't affect lipid homeostasis.

These actions look to be dependent on its coactivator function, as shown by the generation and use of knock-in mice carrying a mutation that specifically impairs the ability of coactivation (K182R). Rescue experiments by using AAV-hEHMT2 overexpression show that the effect on glucose metabolism is liver specific. Although, in the RNA-seq study, the Dex response is only slightly affected (~8% of induced genes), the authors identified *Irs2* and *Dusp4* as possible targets mediating insulin sensitivity. GR and Ehmt2 recruitment seem to be impaired at GBRs of these loci in the mutant mice, and protein expression fails to increase after Dex treatment. Further experiments include the transient overexpression and knockdown of the 2 candidate genes. The experiments show involvement of *Irs2* in the modulation of insulin sensitivity. *Dusp4* activation may modulate the Dex-dependent gluconeogenic program.

Although a link between Ehmt2 and GC regulation of glucose homeostasis appears legitimate, it is not clear if the described functions operate during more physiological conditions. Glucocorticoid hormones are secreted during fasting, and their effect on glucose metabolism is particularly evident at this stage. Can the authors elucidate the role of Ehmt2 during the fasting response (24h and 48h)? It would be also insightful to investigate what happens to the mice during re-feeding following fasting, to test for impairments in insulin action.

We appreciate reviewer's recommendation to study Ehmt2's role in the metabolic regulation of fasting/refeeding switch. We found that the expression of *Irs2* was induced in the liver of wild type mice by 24 hour fasting. This induction was reduced in Ehmt2/K182R mice. Moreover, insulin suppressed glucocorticoid-induced *Irs2* expression. However, *Irs2* expression was also induced by cAMP (glucagon) in hepatocytes (Reference 25, Besse-Patin, A. *et al. PNAS* 2019). We do not know whether Ehmt2 coactivation is involved in cAMP activation of *Irs2* or not. Ehmt2 also interacts with FoxO1, another transcription factor that plays a role in fasting-induced metabolic adaptation. Because Ehmt2 could interact with multiple transcription factors to regulate the fasting/refeeding switch, a systematic approach is needed. Therefore, we believe that this intriguing question is more suited as an independent future project.

A validation at protein level of Dusp4 and Irs2 downregulation in K182R has been done. Since the regulation appears to happen transcriptionally, it is necessary to confirm the downregulation by qRT-PCR in the mouse cohorts. Also, a visual representation of the peak locations used for ChIP-qPCR will be useful.

RT-qPCR data has been added to the figure to show transcriptional regulation of expression (Fig. 4f). The visual representation of the peak locations in mouse genome have been added to the figure (Fig. 4d).

Part of the mechanism is explained by Dusp4 acting on “insulin resistance genes”. The authors show that Dusp4 can indirectly modulate Pck1 and G6pc expression - I found this link to be rather weak, as actually these two genes are not differentially regulated in the mutant mice compared to WT upon DEX. How do the authors explain causality then? Also, what's the protein level of Pck1 and G6pc in Dusp4 KD and OE?

We agree with the reviewer's assessment that the role of Dusp4 in mediating Ehmt2 coactivation on glucocorticoid-regulated insulin sensitivity is minimal. In RNAseq we did not score Pck1 nor G6pc as differentially regulated genes in the liver of wild type and Ehmt2(K182R/K182R) mice. Moreover, hepatic Pck1 and G6pc gene expression under Dex treatment for 11 days was not different between wild type and Ehmt2(K182R/K182R) mice (Supplemental Fig. 1c). Overall, we have revised our model and deleted Dusp4 from the model (Fig. 7).

With respect to the expression of Pck1 and G6pc in Dusp4 KD and OE mice: In Dusp4 overexpressing mice it appears that protein levels of Pck1 and G6pc are decreased (Fig. 5g), but in Dusp4 KD mice Pck1 and G6pc levels are not significantly different (Fig. 5n). These results suggest that while the role of Dusp4 in Ehmt2 coactivation function in Dex response is minimal, altering Dusp4 levels in hepatocytes may modulate gluconeogenic gene and protein expression. This discussion is presented on page 15.

Minor comments:

- The type of DEX treatment in Fig 1 and 3 is assumed to be in drinking water, but is not specified.

This has been added into the text on page 5 and page 8. Additionally we have added this information to the figure legends for Fig 1-3.

- Similarly, the DEX 5h treatment is not described in the methods. Has it been done by IP injections? Which dose?

This has been corrected on page 20 in the methods and in the Figure 4 legend.

- What's the level of expression of mutant K182R protein? Is this comparable to the WT form?

Protein expression of the mutant K182R is similar to WT in mice and we have included this data as Fig. 2c.

- Lane 177 a bracket is missing.

This has been corrected.

- Lane 248, reference to Fig 5i not 5h.

This has been corrected. Because new data has been added it is now lane 262 and is Figure 5i.

- The term 'revolutionary' is absolutely inappropriate in the abstract.

This has been corrected.

Overall, this manuscript includes a lot of in vivo data, and the findings are potentially interesting to the field. However, several shortcomings unfortunately prevent publication in its current form. Most importantly, the evidence that Ehmt2 is a direct GR coactivator (in the molecular sense), and that the effects of Ehmt2 loss of function on GR target gene regulation are direct, need to be solidified and expanded.

Thank you for these comments. In line 62 we cite several papers that have shown Ehmt2 interacts with GR. We have also addressed the concerns by conducting GR, Ehmt2, and Cbx3 ChIP in the liver of WT and Ehmt2 K182R mutant mice. These results demonstrated that Dex increased the recruitment of GR, Ehmt2 and Cbx3 to the Dusp4 and the Irs2 GR binding regions in the liver of wild type mice. The levels of Dex-increased GR and Ehmt2 recruitment to the Dusp4 and the Irs2 GR binding regions were similar between K182R and wild type mice. However, Cbx3 recruitment was significantly decreased for Irs2 and trending to be decreased in Dusp4 GBRs in the liver of Dex treated K182 mutant mice (Fig. 4e). Overall, these results are consistent with the model of Ehmt2 coactivation in which K182 methylation is required for recruiting the coactivator Cbx3.

Reviewer #2 (Remarks to the Author):

The manuscript by Lee et al. examines the mechanism of GR-dependent transcriptional regulation (activation vs repression) in mediating glucocorticoid-induced glucose intolerance and insulin resistance. The authors show that the GR co-regulatory protein, Ehmt2, which is involved in both gene activation and repression, when eliminated in the liver in mice results in more glucose intolerance and insulin resistance upon chronic

glucocorticoid treatment without affecting triglyceride synthesis, thus linking Ehmt2 to glucose homeostasis. To determine whether this is a result of a defect in GR-mediated gene activation vs repression, they engineered a mouse to express a single amino acid substitution in Ehmt2 (K182R) that blocks the co-activator function but retains co-repressor function. The authors use this novel model to demonstrate that this resembles the loss of function of Ehmt2, suggesting that the coactivation function of Ehmt2 is important in controlling glucose homeostasis in mice in vivo. Importantly they show that overexpression of wild type Ehmt2 in the liver of mice in the context of the Ehmt2 K182R mutation restores glucose hemostasis, indicating that the liver is the major tissue involved in the glucose homeostasis by GR and Ehmt2. They go on to identify potential GR targets involved in glucose homeostasis by RNA seq from livers of WT vs Ehmt2 mutant mice. They identify DUSP4 and IRS2 as direct GR targets of GR-Ehmt2 by ChIP. Whereas overexpression of DUSP4 in the liver improved glucose tolerance but not insulin insensitivity, IRS2 overexpression improves both glucose and insulin tolerance in the face of chronic glucocorticoid treatment. Overall the data are compelling and are consistent with the interpretation that the coactivation function of Ehmt2 in the liver is the key determinant of glucocorticoid-induced glucose intolerance and insulin resistance. This represents a novel and important finding.

However there are a few areas of concern. One is that since the Ehmt2 mutant (K182R) mimics Ehmt2 depletion, it begs the question does the Ehmt2 mutant protein (and by association GR) occupy the DUSP4 and IRS2 promoters and fail to coactivate, or does Ehmt2 K182R fail to occupy the DUSP4 and IRS2 regulatory regions resulting in reduced expression? Perhaps this has already been demonstrated by ChIP in the authors previous reports.

This is an important question. We have included ChIP data in the liver of WT and Ehmt2 K182R mutant mice demonstrating that Dex-induced recruitment of Ehmt2 to the Dusp4 and Irs2 GBRs remains intact in K182R mutant mice (Fig. 4e). In the liver of Ehmt2(K182R/K182R) mice, the recruitment of Cbx3 to these GBRs was reduced due to K182R mutation (Fig. 4e).

It would seem from the Ehmt2 complementation study in the liver (which is a very nice experiment) that WT Ehmt2 protein wins out with respect to gene expression, whereas one might posit if the Ehmt2 K182 mutant occupies DNA, then it would act as a dominant negative and block WT EhMT2 activity. However, this does not seem to be the case. It might be worth noting this in the discussion. Is this because WT Ehmt2 is more abundant than Ehmt2 K182 and able to compete effectively to restore gene expression?

Thanks for the thoughtful discussion on this point. While we cannot exactly calculate the relative levels of Ehmt2 and K182R proteins in the liver of K182R mice that were overexpressed with wild type Ehmt2, based on the western blot, total Ehmt2 protein levels were over two fold higher than endogenous K182R proteins (Fig. 3a). Thus, overexpressed wild type Ehmt2 were likely more abundant than endogenous K182R

proteins as the reviewer suggested. This point is included in Page 8, line 175-177.

The effects of IRS2 depletion in the liver by shRNA and subsequent effects on insulin resistance is difficult to discern from Fig 6i? The curves of ITT in the shRNA IRS2 knock down appear superimposable, yet the AUC seem quite different? Not sure how to reconcile this?

We apologize for this error, when transferring the AUC data into the Prism application, one of the data points was left off. When correcting this, we also noticed that the % of baseline calculation for the curves was calculated incorrectly for the 120 minute time point. We have corrected the figure and updated it.

The discussion seems overly speculative: whereas mentioning the GWAS study linking Ehmt2 to type 2 diabetes seems fitting, the speculation about the impact of Ehmt2 T183 phosphorylation seems too hypothetical. That said, the idea that GR might be promoting the effects of insulin is really interesting, and makes us rethink the dogma about how GR effects insulin signaling.

Thanks for this comment. We agree that the role of phosphorylation of T183 in the regulation of GR action is speculative, as we do not have evidence that this phosphorylation occurs in hepatocytes. We have deleted this in the discussion.

Reviewer #3 (Remarks to the Author):

The authors claim that Ehmt2 co-activates Irs2, an insulin-sensitizing gene, along with the glucocorticoid receptor under glucocorticoid treatment. Their data showed that Dex-induced insulin resistance was exacerbated in sh-RNA-treated mice targeting Ehmt2 and Ehmt2(K182R/K182R) mice, and that the latter was ameliorated by overexpression of human EHMT2 or mouse Irs2. Their data showing a correlation between the K182R mutant of Ehmt2 and Dex-induced insulin resistance is interesting and convincing. However, it is still unclear whether the K182R mutation is directly related to the activation of GR-dependent transcription. In addition, considering the site-specific regulations of transcription by histone modifications, the immunoblot analysis of liver tissue extract using anti-H3K9me2 is insufficient to conclude that “K182R mutation does not affect the methyltransferase activity of mouse Ehmt2 in vivo”. Therefore, the authors should conduct additional experiments to clarify the mechanism by which Ehmt2 mediates Dex-induced insulin resistance.

1. It is unclear whether the co-activation of GR target genes by Ehmt2 occurs by a mechanism independent of histone H3K9 methylation. In Figure 4d, ChIP-qPCR should be performed in both WT and Ehmt2(K182R/K182R) mice using the anti-H3K9me2 antibody in addition to anti-GR and anti-Ehmt2 antibodies. Furthermore, it is not clear whether Ehmt2 is recruited by GR and methylated to activate gene expression in mouse liver as reported in the previous study used cell culture models by Poulard et al. (Ref.

#12). It should be confirmed whether Cbx3 is recruited to GR target genes in mouse liver in a methylated Ehmt2-dependent manner. If not, what is the mechanism by which GR is activated by methylated Ehmt2 in mouse liver?

Thank you for this note.

We performed ChIP of H3K9me2 in the liver of wild type, Ehmt2(K182R/K182R) mice and hepatic Ehmt2 knockdown mice. A previous study has performed a whole genome H3K9me2 ChIP sequencing in mouse liver. One region that contains significant levels of H3K9me2 is chromosome 3 (mm9 chr3:56,379,000-56,380,000). We found that H3K9me2 levels in this region were similar between WT and *Ehmt2*^{K182R/K182R} mice (Supplemental Figure 1a). But, H3K9me2 levels in this region were lower in Ehmt2 knockdown mouse liver (Supplemental Figure 1a). These results were described in line 130-136.

We also included the results of Ehmt2 and Cbx3 ChIP data. Dex treatment induced the similar levels of Ehmt2 to the *Dusp4* and the *Irs2* GBRs in wild type and Ehmt2(K182R/K182R) mice (Fig. 4e). However, Dex-induced Cbx3 recruitment to the *Dusp4* and the *Irs2* GBRs was reduced in Ehmt2(K182R/K182R) mice (Fig. 4e).

2. The manuscript on page 12 does not match the numbering of the figures (Figure 5h and following). In some places, it is not possible to guess which part of the figure is being discussed, which makes it difficult to evaluate the content.

This has been corrected. The shRNA targeting *Dusp4* data appears line 257-274 and the figure numbering is Figure 5h-n.

3. The intensity of the immunoblot analysis of *Irs2* in GA and eWAT in Fig. 6f is too weak to confirm that there is no significant reduction by sh-*Irs2*. mRNA expression should alternatively be presented.

We have done gene expression and have included it in Fig. 6f.

4. In relation to the regulation of hepatic gluconeogenesis by *Dusp4*, the authors should present the expression of the gluconeogenic genes in the liver of Ehmt2(K182R/K182R) mice.

In RNAseq (5hr Dex treatment), we did not score either *Pck1* or *G6pc* as a differentially regulated gene in the liver of Ehmt2(K182R/K182R) mice. For a longer Dex treatment (11 days), we also did not find the differential regulation of these two genes in the liver of wild type and Ehmt2(K182R/K182R) mice (Supplemental Figure 1c). We state these results on page 15 line 382. These results and the fact that *Dusp4* knockdown in mice liver did not recapitulate the phenotype of Ehmt2(K182R/K182R) mice indicate that *Dusp4* plays a modest role in Dex response in Ehmt2(K182R/K182R) mice. This discussion is presented in line 351-361. We also deleted *Dusp4* from the model (Fig. 7)

because its role in mediating Ehm2 coactivation regulated Dex-induced insulin resistance is modest.

REVIEWER COMMENTS

Reviewer #1 (Remarks to the Author):

In this revised version of the manuscript from the Wang lab, the authors have done their best to address my comments and concerns.

However, I am still not convinced that their data support their claims, unfortunately.

My major concern regards the two GR targets, with the conclusion that Dusp4 regulates Pck1 and G6pc. The reported effect on gene expression is minimal and the WBs using GAPDH as loading control are in appropriate (GAPDH is a GR target and known to change, especially with regards to hepatic metabolism):

In fig 5g, the GAPDH level is increasing, which the authors take to state that Pepck is down in the overexpressor. I fail to see that the levels of Pepck or G6pase change reproducibly.

Since Pck1 and G6pc are gluconeogenic genes, why they haven't performed a PTT?

The paper's biggest shortcoming is really figure 5, which in my opinion shows that Pck1 and G6pc do not change in their model (ehmt2 knock-in), so there is no mechanistic link.

Also, the GTTs (fig 5 and fig 6) are puzzling: the respective controls for overexpression and silencing, GFP and scramble, show a different glucose tolerance under DEX. Is there some off-target effect? In both experiments, there is an increase of insulin in Dex treated mice (which is normal), but the overall effect on glucose tolerance is different between the groups.

Taken together, I unfortunately cannot endorse publication in Nat Comms.

Reviewer #2 (Remarks to the Author):

The authors have effectively addressed the concerns raised in the prior review.

Reviewer #3 (Remarks to the Author):

The authors have adequately addressed my previous comments. In my view, this manuscript is suitable for publication in Nature Communications.

We appreciate reviewer 1's comments. Below is our response and revision:

Reviewer #1 (Remarks to the Author):

In this revised version of the manuscript from the Wang lab, the authors have done their best to address my comments and concerns.

However, I am still not convinced that their data support their claims, unfortunately.

My major concern regards the two GR targets, with the conclusion that Dusp4 regulates Pck1 and G6pc. The reported effect on gene expression is minimal and the WBs using GAPDH as loading control are in appropriate (GAPDH is a GR target and known to change, especially with regards to hepatic metabolism):

In fig 5g, the GAPDH level is increasing, which the authors take to state that Pepck is down in the overexpressor. I fail to see that the levels of Pepck or G6pase change reproducibly.

Since Pck1 and G6pc are gluconeogenic genes, why they haven't performed a PTT?

The paper's biggest shortcoming is really figure 5, which in my opinion shows that Pck1 and G6pc do not change in their model (ehmt2 knock-in), so there is no mechanistic link.

Thank you for pointing out the potential issue of using GAPDH an internal control in immunoblotting. We have now used tubulin as an internal control in this revised manuscript and our conclusion is that there is no difference in Pck1 and G6pc protein levels after Dusp4 overexpression (new Fig. 5g). Thus, neither overexpressing nor reducing Dusp4 affects Pck1 and G6pc protein expression. Therefore, the conclusion is that Dusp4 does not mediate the Ehmt2 K182R phenotype. This message is clearly presented in the manuscript (page 11, line 258-9). Notably, we did not include Dusp4 in the model (Fig. 7). Thus, these results do not affect our model.

We performed a PTT in hepatic Dusp4 knockdown mice that were treated with Dex. We did not observe any difference of gluconeogenic capacity between Dex treated control and Dusp4 knockdown mice (Supplemental Figure 2). This agrees with the protein expression results. These results are presented in the manuscript (line 274-277).

Also, the GTTs (fig 5 and fig 6) are puzzling: the respective controls for overexpression and silencing, GFP and scramble, show a different glucose tolerance under DEX. Is there some off-target effect? In both experiments, there is an increase of insulin in Dex treated mice (which is normal), but the overall effect on glucose tolerance is different between the groups.

Thanks for pointing this out. Indeed, we consistently observed that dexamethasone (Dex) treated hepatic GFP overexpression mice were glucose intolerant (Fig. 5b and Fig. 6b), but Dex treated hepatic scramble (Scr) shRNA expressing mice had improved glucose tolerance (Fig. 5i and Fig. 6g). The exact reason for these differences is unknown. The shRNA experiments were performed in WT C57BL/6J mice while the overexpression experiments were performed in the Ehmt2 K182R mice. To make this easier to interpret in the figures, we have adjusted the color scheme such that experiments using WT mice are in Black/Red color schemes and the experiments using Ehmt2 K182 mice are in Pink/Blue color schemes. In any case, Dex treatment caused insulin resistance (based on ITT) in both GFP overexpression mice (Fig. 5d and 6d) and Scr shRNA overexpression mice (Fig. 5k and 6i). Similarly, Dex treatment caused hyperinsulinemia in both GFP overexpression mice (Fig. 5c and 6c) and Scr shRNA overexpression mice (Fig. 5j and 6h). Thus, the different glucose tolerance in GFP and Scr shRNA overexpression experiments could be due to the varying degrees of insulin resistance.

In Scr shRNA overexpression mice, Dex-induced hyperinsulinemia not only normalizes, but in fact reduces blood glucose levels. However, in GFP overexpression Ehmt2 K182R mice, Dex treatment induced hyperinsulinemia is not able to normalize the blood glucose levels. Overall, Dex-induced insulin resistance (ITT) and hyperinsulinemia have been very consistent. So, the discrepancy of the GTT results in GFP and Scr shRNA overexpressing mice does not affect the conclusion of our model.

REVIEWERS' COMMENTS

Reviewer #4 (Remarks to the Author):

To the authors:

This manuscript brings interesting new insights into the specific role of Ehmt2 as a coactivator of GR in the context of GC-induced insulin resistance. It casts light on the future potential of improving GC therapeutic strategies by manipulating GR coactivators to finetune treatments and minimizing side-effects including insulin-resistance. The revised version of the manuscript and the response to concerns about Figure 5 and the color changes in Figure 6 have to my opinion addressed the concerns and I'm satisfied with the current version of the manuscript.

I have some minor comments and questions that you might want to consider by rephrasing or including to clarify and further support of the conclusions, but they are not strictly necessary, and I will recommend this manuscript being considered for publication regardless.

Questions and comments:

Line 31: In the abstract you state that you find that GR stimulates the transcription of *Irs2*: "Here, we found that GR also stimulates the transcription of insulin-sensitizing genes, such as *Irs2*". Using this phrasing in the abstract I had expected to see some data showing that GC-induced *Irs2* mRNA expression requires functional GR expression. E.g. an experiment disrupting GR expression/function in combination with dex stimulation. You do indeed show that GC stimulates *Irs2* and *Dusp4* and that there are GR binding to chromatin near the genes, but it could in principle be unrelated. GC could mediate its effects on *Irs2* and *Dusp4* expression through non-genomic GC actions, secondary effects or through e.g. the progesterone receptor which has been suggested to be activated by high levels of GC (PMID: 14617569) and a regulator of *Irs2* (PMID: 10077005). I agree that the GC effects in this case is most likely through GR, but I don't agree that your results show that. The same applies in Figure 4 legend, line 313 and 403 where you write that *Irs2* and *Dusp4* are "GR primary target genes". I suggest that you refer to publications showing that *Dusp4* and *Irs2* are primary GR target genes or rephrase.

Line 63-65: Just to set the scene for the reader: would it be relevant to mention the whole coactivator complex with GRIP1, CARM1 and p300 and the importance of this lysine 182 for the whole coactivator complex function?

Line 180-181: You state that the overexpression of Ehmt2 improved glucose tolerance in mice without dex, but in the Figure 3b it is "ns". So, it is more trending towards improved glucose tolerance?

Line 209: Here you mention that 677 genes are induced by dex. I suggest that you list just a few examples of classical GR target genes that you find among these either in the text or in Figure 4 – just to show that the mice responded as expected to the acute dex treatment. Alternatively, or additionally, consider including a table in Supplementary file 1 or in an extra column in the Supplementary table 1 listing those 677 dex-induced genes highlighting the 57 genes reduced in the mutant so the readers can

easily check the genes you find.

Line 220: It is not clear why you chose those to focus on those 2 specific GR binding sites (GRBS) for *Irs2* and *Dusp4*. Going into the data from the reference #27, where I believe you got the GRBS coordinates from, I see that e.g. for *Irs2* there are other GRBSs around (*Irs2* +34.1kb and *Irs2* -12.8kb) which are closer to the *Irs2* promoter than the one you went for (~-42.6 kb), so I guess your selection was not only based on proximity? Maybe you have found, or it has been reported, that the *Irs2* and *Dusp4* GRBSs you selected are regulatory region of these genes in e.g. luciferase reporter assays or CRISPR experiments? Or maybe available chromatin conformation data has showed interaction between these sites and the promoter? This information would also support the model you have in Figure 7 where the green arrow suggesting an interaction.

Please include the genomic location of the GRBS you have decided to analyze either in the text, in Figure 4e figure heading or Figure 4 legend. This could also be included in the primer table.

Please also mention if the GRBS you have selected are based on GR ChIP-seq data, which I assume is the case based on reference#27, or if you have selected the sites based on e.g. GRE motifs.

Line 231: You refer to Figure 4f saying that it shows gene expression data, but I see a Western? Was the gene expression data accidentally removed from Figure 4? I see in your source data file that where should be gene expression data.

Methods section: You mention in the main text that you use male mice, but it is not clear if you used males for all your experiments. I suggest that you make that clear in the method section along with the age of the mice when glucose and insulin tolerance test were performed.

Methods section: Please provide information about the pyruvate tolerance test.

Methods section: If possible, add catalog numbers on the Adenovirus and AAV you have purchased from Vector Biolabs.

Typos and phrasing:

Line 175 + Figure legend 5 and 6: You write "GFP overexpressing mice". I would only call something an overexpression if the protein in question is expressed to some extent at basal level, which is not the case for GFP. I suggest you change it to "GFP expressing mice".

Line 179: typing error: "pf"  "of".

Line 201-202: In Supplementary file 1, the figure legend for this figure is called "S1" and not "S2" as it says in the main text.

Line 207-208: This sentence is not clear to me. What previous work and what kind of upregulation are you referring to here?

In Figure 1, 2, 3 and 4 legends you have included description of the significance stars and the error bars, but this is not done in Figure 5 and 6 legends. I suggest that you are consistent.

Figure 1 legend: The figure title states that hepatic Ehmt2 KD doesn't affect hypertriglyceridemia, but you don't show it in the figure. I would suggest you change the title of the figure so that it reflects what you show in that figure or in a supplementary figure supporting that figure.

Figure 2 legend: "PGTT"  "IPGTT".

Supplementary Figure 1 legend: "(...) . N=3-4"  "(...), n=3-4".

Response to the reviewer 4:

We appreciate the reviewer's comments. We have revised the manuscript based on the reviewer's comments. In the revised manuscript, all revisions are highlighted by yellow.

To the authors:

This manuscript brings interesting new insights into the specific role of Ehmt2 as a coactivator of GR in the context of GC-induced insulin resistance. It casts light on the future potential of improving GC therapeutic strategies by manipulating GR coactivators to finetune treatments and minimizing side-effects including insulin-resistance. The revised version of the manuscript and the response to concerns about Figure 5 and the color changes in Figure 6 have to my opinion addressed the concerns and I'm satisfied with the current version of the manuscript.

I have some minor comments and questions that you might want to consider by rephrasing or including to clarify and further support of the conclusions, but they are not strictly necessary, and I will recommend this manuscript being considered for publication regardless.

Questions and comments:

Line 31: In the abstract you state that you find that GR stimulates the transcription of Irs2: "Here, we found that GR also stimulates the transcription of insulin-sensitizing genes, such as Irs2". Using this phrasing in the abstract I had expected to see some data showing that GC-induced Irs2 mRNA expression requires functional GR expression. E.g. an experiment disrupting GR expression/function in combination with dex stimulation. You do indeed show that GC stimulates Irs2 and Dusp4 and that there are GR binding to chromatin near the genes, but it could in principle be unrelated. GC could mediate its effects on Irs2 and Dusp4 expression through non-genomic GC actions, secondary effects or through e.g. the progesterone receptor which has been suggested to be activated by high levels of GC (PMID: 14617569) and a regulator of Irs2 (PMID: 10077005). I agree that the GC effects in this case is most likely through GR, but I don't agree that your results show that. The same applies in Figure 4 legend, line 313 and 403 where you write that Irs2 and Dusp4 are "GR primary target genes". I suggest that you refer to publications showing that Dusp4 and Irs2 are primary GR target genes or rephrase.

Thank you for the reviewer's comment. We understand the reviewer's point and revised the term "GR primary target genes" to "potential GR primary target genes" or "glucocorticoid-regulated genes". In line 31, we rephrased "GR also stimulates the transcription of insulin-sensitizing genes" to "glucocorticoids also stimulate the expression of insulin-sensitizing genes". Notably, the line numbers here are based on the revised manuscript. In line 188, we revised "EHMT2 coactivation function dependent GR primary target genes" to "EHMT2 coactivation function-dependent potential GR primary target genes". In line 328 we revised "GR primary target genes" to "potential GR primary target genes". In lines 419-420 we revised "GR primary target genes" to "glucocorticoid regulated genes".

Line 63-65: Just to set the scene for the reader: would it be relevant to mention the whole coactivator complex with GRIP1, CARM1 and p300 and the importance of this lysine 182 for the whole coactivator complex function?

Thank you for this comment. We have added a sentence to highlight the interaction between Ehmt2 and these coactivators in line 60-61 of the revised manuscript.

Line 180-181: You state that the overexpression of Ehmt2 improved glucose tolerance in mice without dex, but in the Figure 3b it is "ns". So, it is more trending towards improved glucose tolerance?

Thanks for pointing this out. We have revised this in the figure 3b. The p value is 0.02 not ns.

Line 209: Here you mention that 677 genes are induced by dex. I suggest that you list just a few examples of classical GR target genes that you find among these either in the text or in Figure 4 – just to show that the mice responded as expected to the acute dex treatment. Alternatively, or additionally, consider including a table in Supplementary file 1 or in an extra column in the Supplementary table 1 listing those 677 dex-induced genes highlighting the 57 genes reduced in the mutant so the readers can easily check the genes you find.

We have added a sentence to list some previously identified GR regulated genes (line 214-215). All 57 genes are highlighted by yellow in supplementary file 2. This is indicated in line 516-518.

Line 220: It is not clear why you chose those to focus on those 2 specific GR binding sites (GRBS) for *Irs2* and *Dusp4*. Going into the data from the reference #27, where I believe you got the GRBS coordinates from, I see that e.g. for *Irs2* there are other GRBSs around (*Irs2* +34.1kb and *Irs2* -12.8kb) which are closer to the *Irs2* promoter than the one you went for (~-42.6 kb), so I guess your selection was not only based on proximity? Maybe you have found, or it has been reported, that the *Irs2* and *Dusp4* GRBSs you selected are regulatory region of these genes in e.g. luciferase reporter assays or CRISPR experiments? Or maybe available chromatin conformation data has showed interaction between these sites and the promoter? This information would also support the model you have in Figure 7 where the green arrow suggesting an interaction.

Please include the genomic location of the GRBS you have decided to analyze either in the text, in Figure 4e figure heading or Figure 4 legend. This could also be included in the primer table. Please also mention if the GRBS you have selected are based on GR CHIP-seq data, which I assume is the case based on reference#27, or if you have selected the sites based on e.g. GRE motifs.

The reason we focused on the -42.7kb GBR of *Irs2* gene is because in the conventional CHIP, this GBR showed a clear and consistent recruitment of GR. In the meantime, for the GBR of *Irs2* gene at -34.1 kb we are not able to confirm GR recruitment. This point is presented in the manuscript (line 228-234).

Line 231: You refer to Figure 4f saying that it shows gene expression data, but I see a Western? Was the gene expression data accidentally removed from Figure 4? I see in your source data file that where should be gene expression data.

Thanks for pointing this out. We have added gene expression results for Figure 4f.

Methods section: You mention in the main text that you use male mice, but it is not clear if you used males for all your experiments. I suggest that you make that clear in the method section along with the age of the mice when glucose and insulin tolerance test were performed.

Methods section: Please provide information about the pyruvate tolerance test.

This has been added.

Methods section: If possible, add catalog numbers on the Adenovirus and AAV you have purchased from Vector Biolabs.

This information is added.

Typos and phrasing:

Line 175 + Figure legend 5 and 6: You write “GFP overexpressing mice”. I would only call something an overexpression if the protein in question is expressed to some extent at basal level, which is not the case for GFP. I suggest you change it to “GFP expressing mice”.

This has been changed (line 179).

Line 179: typing error: “pf”  “of”.

This has been corrected (line 183).

Line 201-202: In Supplementary file 1, the figure legend for this figure is called “S1” and not “S2” as it says in the main text.

This has been corrected.

Line 207-208: This sentence is not clear to me. What previous work and what kind of upregulation are you referring to here?

This sentence is revised (line 212-213 of revised manuscript). The reference for this statement is added.

In Figure 1, 2, 3 and 4 legends you have included description of the significance stars and the error bars, but this is not done in Figure 5 and 6 legends. I suggest that you are consistent.

This has been included.

Figure 1 legend: The figure title states that hepatic Ehmt2 KD doesn't affect hypertriglyceridemia, but you don't show it in the figure. I would suggest you change the title of the figure so that it reflects what you show in that figure or in a supplementary figure supporting that figure.

This has been corrected.

Figure 2 legend: “PGTT”  “IPGTT”.

This has been corrected.

Supplementary Figure 1 legend: “(...) . N=3-4”  “(...), n=3-4”.

This has been corrected.